# A new polymodal gating model of the proton-activated chloride channel

**Piao Zhao**[1,2☯], **Cheng Tang**[1,2☯]*, **Yuqin Yang**[3☯], **Zhen Xiao**[1], **Samantha Perez-Miller**[4], **Heng Zhang**[5], **Guoqing Luo**[1], **Hao Liu**[1], **Yaqi Li**[1], **Qingyi Liao**[1], **Fan Yang**[5], **Hao Dong**[3]*, **Rajesh Khanna**[4,6]*, **Zhonghua Liu**[1,2]*

**1** The National and Local Joint Engineering Laboratory of Animal Peptide Drug Development, College of Life Sciences, Hunan Normal University, Changsha, China, **2** Peptide and small molecule drug R&D platform, Furong Laboratory, Hunan Normal University, Changsha, China, **3** Kuang Yaming Honors School, State Key Laboratory of Analytical Chemistry for Life Science, Engineering Research Center of Protein and Peptide Medicine of Ministry of Education, & Institute for Brain Sciences, Nanjing University, Nanjing, Jiangsu, China, **4** Department of Molecular Pathobiology and NYU Pain Research Center, College of Dentistry, New York University, New York, New York, United States of America, **5** Department of Biophysics and Kidney Disease Center, The First Affiliated Hospital, Zhejiang University School of Medicine, Hangzhou, Zhejiang, China, **6** Department of Neuroscience and Physiology and Neuroscience Institute, School of Medicine, New York University, New York, New York, United States of America

☯ These authors contributed equally to this work.
* chengtang@hunnu.edu.cn (CT); donghao@nju.edu.cn (HD); rk4272@nyu.edu (RK); liuzh@hunnu.edu.cn (ZL)

**Data Availability Statement:** The authors confirm that all data underlying the findings are fully available without restriction. All relevant data are within the paper and its Supporting Information files.

## Abstract

The proton–activated chloride (PAC) channel plays critical roles in ischemic neuron death, but its activation mechanisms remain elusive. Here, we investigated the gating of PAC channels using its novel bifunctional modulator C77304. C77304 acted as a weak activator of the PAC channel, causing moderate activation by acting on its proton gating. However, at higher concentrations, C77304 acted as a weak inhibitor, suppressing channel activity. This dual function was achieved by interacting with 2 modulatory sites of the channel, each with different affinities and dependencies on the channel's state. Moreover, we discovered a protonation–independent voltage activation of the PAC channel that appears to operate through an ion–flux gating mechanism. Through scanning–mutagenesis and molecular dynamics simulation, we confirmed that E181, E257, and E261 in the human PAC channel serve as primary proton sensors, as their alanine mutations eliminated the channel's proton gating while sparing the voltage–dependent gating. This proton–sensing mechanism was conserved among orthologous PAC channels from different species. Collectively, our data unveils the polymodal gating and proton–sensing mechanisms in the PAC channel that may inspire potential drug development.

## Introduction

Protons, possibly the smallest ligands in biology, regulate the activity of various ion channels on plasma and intracellular membranes by modifying the channels' response to other gating stimuli [1–4], or by acting as direct channel agonists [5], or both [6]. While the proton–activated, acid–sensing cation channels (ASICs) were identified decades ago and have been

**Funding:** This work was supported by the National Natural Science Foundation of China (31600669 and 32171271 to C. T., 32071262 and 31770832 to Z. L., and 22273034 to H. D.), the Science and Technology Innovation Program of Hunan Province (2020RC4023 to Z. L.), the Natural Science Foundation of Hunan Province (2018JJ3339 to C. T.), the Research Foundation of the Education Department of Hunan Province (18B015 and 22A0076 to C. T.), and the Fundamental Research Funds for the Central Universities (021514380018 to H. D.). Parts of the calculations were performed using computational resources on an IBM Blade cluster system from the High-performance Computing Center (HPCC) of Nanjing University. The funders had no role in study design, data collection and analysis, decision to publish, or preparation of the manuscript.

**Competing interests:** The authors have declared that no competing interests exist.

**Abbreviations:** ASIC, acid–sensing cation channel; ASOR, acid–sensing, outwardly rectifying; DMEM, Dulbecco's Modified Eagle Medium; ECD, extracellular domain; ECL, extracellular loop; EGFP, enhanced green fluorescense protein; PAC, proton–activated chloride; PD, pore domain; PS, pregnenolone sulfate; wt, wild–type.

intensively studied [7], the proton–activated chloride (PAC) channel contributing to acid–sensing, outwardly rectifying (ASOR) anion currents, was only recently characterized [8–10]. The PAC channel is a homotrimer, with each subunit composed of 2 transmembrane segments (TM1 and TM2) connected by a spanning extracellular loop (ECL) [8,9,11–13]. Spatially, the pore–lining helix TM2 and the peripheral TM1 create the pore domain (PD), while the ECL forms the extracellular domain (ECD); the ECD constricts at the extracellular membrane face to join the PD [11–13]. A critical lysine residue at the intracellular end of TM2 serves as the Cl⁻selectivity filter [11]. Both the topological and spatial structures of the PAC channel bear similarity to ASICs and the ATP–gated purinergic P2X receptor cation channels [14,15]. The large, hand–like ECD domain in the PAC channel likely contains multiple sites for pharmacological modulation, as that in ASICs [16]. However, unlike ASICs, the ECD of the PAC channel lacks the exterior helix domain that comprises the presumptive acid pocket, suggesting that it may work via a specialized proton–sensing mechanism [11].

Structural biology approaches have offered insights into possible mechanisms of proton sensing in PAC. Comparison of the resting and desensitized PAC structures implicated a histidine residue at the extracellular end of TM1 (H98 –human PAC numbering) as the proton sensor, with S102, Q296, and I298 forming its docking site (pH 8.0); in the protonated state (pH 4.0), the sensor is believed to reside in the "acid pocket" formed by E107, D109, and E250 [11]. However, mutating these residues did not attenuate the channel's proton sensitivity [11], raising questions as to their roles in proton sensing. Translating the differences between the resting and the putative desensitized, but not the truly opened, PAC channel structures to conformational changes initiated by acid sensing might be problematic. Comparing the resting and activated PAC channel structures, the Long group proposed another proton–sensing mechanism, in which 6 titratable residues likely create the proton sensor and form pairwise interactions in the channel's activated state (E257–D289, E249–E107, and E250–D297) [13]. However, it is unknown whether such interactions are the initial driving force for proton gating. Consequently, the proton–sensing mechanism in the PAC channel remains poorly understood, as that in ASICs [17]. The gating mechanism of PAC channel could be much more complicated than that of ASICs, for which a depolarization–facilitated gating was reported [18]. Since the PAC channel does not possess a prototypical voltage sensor commonly found in classic voltage–gated ion channels [19], they may sense voltage via an unorthodox mechanism.

As an evolutionarily conserved protein [8,9], the PAC channel plays critical physiological and pathological roles. Reducing or abolishing PAC activity protects cells from acid–induced necrotic cell death in vitro and in animal models [8,20,21]. The PAC channel has also been reported to function as a pH sensor to prevent hyper–acidosis in endosomes [22]. Therefore, pharmacological agents modulating PAC activity can be leveraged as molecular probes to investigate key structure–function relationships. Unfortunately, only nonspecific PAC channel antagonists abound, and their mechanism of action remains unknown [21,23–25].

Herein, we explored PAC channel gating using its novel and unique bifunctional modulator C77304 (5–iodo–2–(2–methylfuran–3–carboxamido)benzoicacid) as a pharmacological tool. By uncovering C77304's mechanism of action, we demonstrated that the PAC channel undergoes protonation–independent voltage gating. Scanning–mutation analysis and molecular dynamics simulations revealed that the E181, E257, and E261 mutations in human PAC specifically abolished the channel's proton gating but spared its voltage–dependent gating. Cross–species analysis confirmed that this proton–sensing mechanism is both conserved and variable among orthologous PAC channels. Altogether, our data advances foundational knowledge of the gating mechanism of PAC channel, which may spur interest in targeting it for drug development.

## Results

### C77304 bifunctionally modulates the PAC channel

Pharmacological agents interfering with the proton gating of the PAC channel are valuable molecular tools for dissecting its gating mechanism. We screened a compound library (from Selleck Chemicals LLC, Catalog No. L3600) for such agents using manual patch–clamp analysis. Specifically, a total of 4,208 chemical compounds featuring different core structures and structural diversities from this library were tested (at 10 μM) for their ability to inhibit pH 4.6–evoked PAC currents in HEK293T cells (currents recorded by ramp depolarizations from −70 to +80 mV). Compounds demonstrating >50% inhibition of the pH 4.6–evoked PAC currents were considered as positive "hits" (3 independent assays per compound), which were further tested at a less acidic pH of 5.34 to evaluate any possible pH–dependence of the inhibition. Four hit molecules, designated compounds 77304, 65841, 62975, and 66386 [C77304, C65841, C62975, and C66386; Fig 1A (upper panel), Figs 1B and S1A–S1C], were identified from this screening campaign. Unexpectedly, C77304 but not the other 3 (at 10 μM) exhibited activation effects on the pH 5.34–evoked PAC currents, identifying it as a unique bifunctional modulator and suggested that it might have a high likelihood of interacting with the channel's proton gating (Figs 1C and S1A–S1C). We thus selected and synthesized C77304 for further investigation (S1D–S1G Fig).

A higher concentration of C77304 (at 50 μM) resulted in similar inhibitory effect on both the pH 4.6– and pH 5.34–evoked PAC currents (Fig 1B and 1D). These findings indicate that C77304 exhibits distinct modulatory effects on the macroscopic PAC currents at the same pH of 5.34, depending on the applied concentrations (refer to Fig 1C and 1D). The activation effect of C77304 on PAC currents at pH 5.34, as opposed to the inhibition at pH 4.6, cannot be attributed to pH fluctuations caused by the compound's acid dissociation (lower panel in Fig 1A), nor to changes in the compound's deprotonation state, which remained stable at pHs ranging from 4.6 to 7.4 as determined by computational analysis (S1H Fig). Instead, it is likely influenced by the varying ratios of closed and open state channels observed under these 2 pH conditions. Further investigation revealed that under weak acidic conditions, high concentrations of C77304 (e.g., pH 5.6, 50 μM compound) exhibited a time–dependent effect on PAC currents. Specifically, the currents demonstrated transient activation at the initial time point, but gradually became inhibited thereafter (Fig 1E). However, pretreatment with 1 μM C77304, to stably potentiate PAC channel opening that consequently up–regulates the open state channel proportion, abolished the transient activation event and now the currents were directly inhibited by the compound (Fig 1F). These findings suggest that the ratio of closed and open state PAC channels on the membrane, along with the concentration of C77304, jointly determined both the initial and steady–state effects of C77304 on the macroscopic PAC currents.

Indeed, the concentration–response curves of C77304 against the PAC channel were bell–shaped under relatively weak acidic conditions (i.e., pH 5.34, pH 5.6, pH 5.8), demonstrating bifunctional modulation (Fig 1G). When the pH was below 5.0, C77304 consistently inhibited PAC currents because most of the channels were in the open state (Fig 1G). It is worth noting that, at pH 5.6, the channels activated by both protons and the compound accounted for only a small portion of the total channels that could be activated. This was determined by comparing the currents to those evoked by pH 4.6 and normalizing them accordingly (Fig 1H). The divergence between the activation and inhibition phenotypes is likely because the $EC_{50}$ for C77304 activating the PAC channel was roughly 9– to 25–fold lower than its $IC_{50}$ in blocking it (Fig 1I and 1J). Moreover, the inhibitory effect of C77304 on PAC channels was slightly attenuated under conditions where a bifunctional modulation was observed (Fig 1I), suggesting that the compound exhibited different potencies and/or efficacies in inhibiting the proton–and

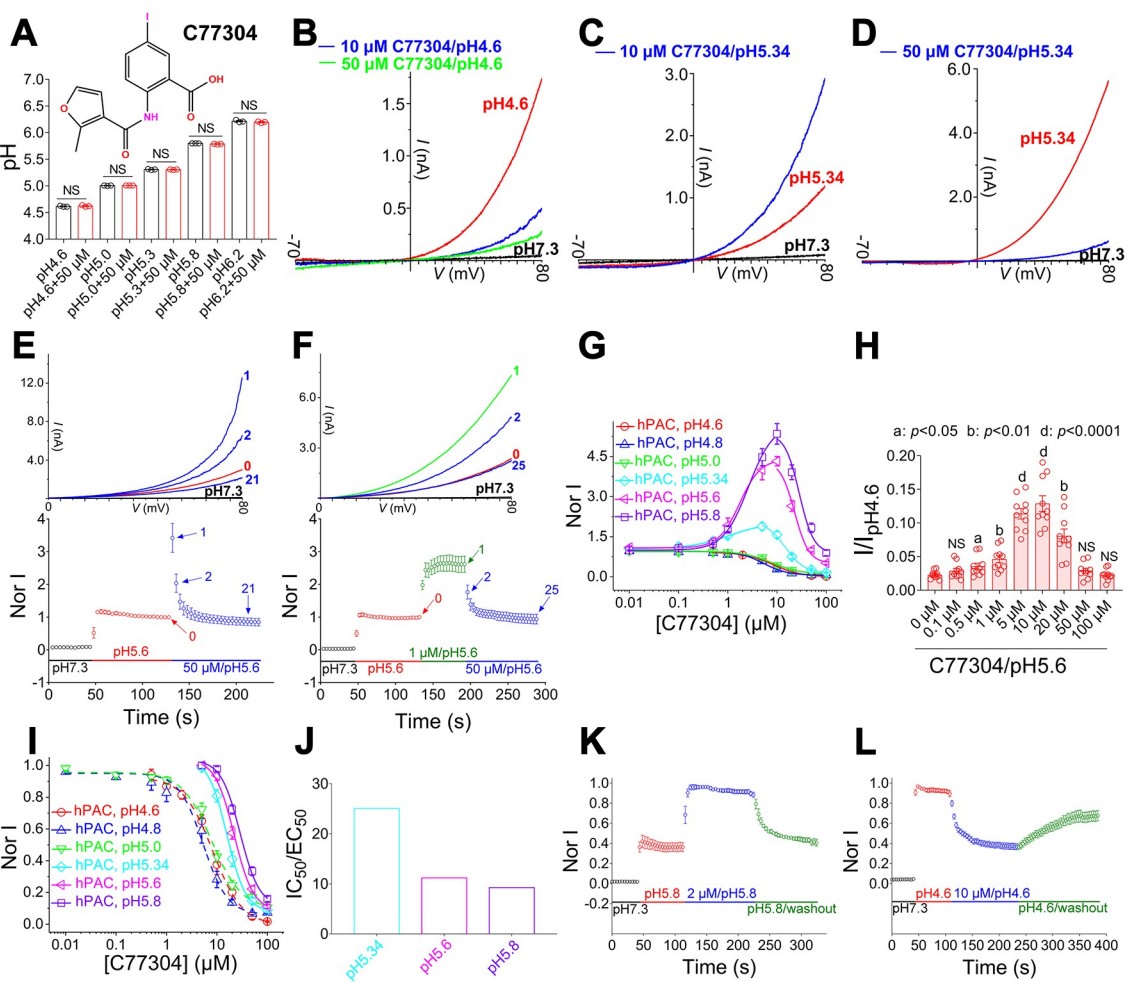

**Fig 1. Identification of C77304 as a bifunctional modulator of the PAC channel.** (A) Upper panel, C77304 structure; lower panel, C77304 does not change buffer pHs ($n = 3$; NS, not significant; paired $t$ test). (B) C77304 concentration–dependently inhibits pH 4.6–evoked PAC currents recorded at ramp depolarizations from −70 to +80 mV (1 mV/ms) ($n = 6$). (C, D) C77304 at 10 μM potentiates (C), but at 50 μM inhibits (D), PAC currents at pH 5.34 ($n = 15$). (E, F) Lower panels: time–course of C77304 acting on PAC channels at pH 5.6. Colored bars show the treatment sequence. Currents were normalized to time point 0. Upper panels: example current traces at different time points as indicated ($n = 7–10$). (G) Concentration–response curves of C77304 acting on PAC channels at different pHs. Currents were normalized to the no compound treatment condition. The activation EC$_{50}$s and mean hill slopes were 2.9 ± 0.9 μM and 1.6 ($n = 19$), 1.8 ± 0.6 μM and 1.9 ($n = 8$), and 0.6 ± 0.2 μM and 2.1 ($n = 16$) at pH 5.8, pH 5.6, and pH 5.34, respectively. (H) Concentration–response relationship of C77304 acting on PAC channels at pH 5.6, currents were normalized to the maximum available current elicited by pH 4.6 stimulus; open circles in each bar represents a separate experimental cell [$n = 10$; significant differences between the C77304 treated groups and the control group (0 μM) were assessed using RM (repeated measure) one–way ANOVA with post hoc Dunnett analysis; NS, not significant]. (I) The same as in (G), except that currents were normalized to the maximum compound–activated current (5 or 10 μM) at pHs 5.34, 5.6, and 5.8, the IC$_{50}$s and mean hill slopes were 26.9 ± 2.6 μM and 1.9 ($n = 19$), 20.2 ± 1.5 μM and 2.5 ($n = 8$), 15.5 ± 1.1 μM and 2.6 ($n = 15$), 7.9 ± 0.5 μM and 2.0 ($n = 8$), 6.2 ± 0.8 μM and 1.8 ($n = 7$), and 8.3 ± 1.0 μM and 1.5 ($n = 6$) at pH 5.8, pH 5.6, pH 5.34, pH 5.0, pH 4.8, and pH 4.6, respectively. (J) The ratio of the apparent affinity of C77304 inhibiting and activating PAC channels (IC$_{50}$/EC$_{50}$) at different pHs. (K, L) Time–course of C77304 activating (K) and inhibiting (L) PAC currents and the corresponding wash–out upon perfusion with compound–free solutions. Compound activation and subsequent current recovery at pH 5.8 had time constants of 4.5 ± 1.3 s and 15.9 ± 2.5 s, respectively. At pH 4.6, compound inhibition and subsequent current recovery had time constants of 15.9 ± 2.3 s and 82.7 ± 12.9 s, respectively ($n = 6–10$). The data underlying the graphs shown in the figure can be found in S1 Data. PAC, proton–activated chloride.

compound–activated channels compared to the proton–activated channels. The compound's activation and inhibition effects remained stable after their fast onset upon acute perfusion (Fig 1K and 1L). Additionally, it was observed that the activation effect was fully reversible while the inhibition effect was only partially reversible (Fig 1K and 1L). Altogether, it is likely

that the effect of C77304 on PAC channels is driven by a competition between its activating and inhibiting actions.

## The mechanisms of C77304 bifunctionally modulating the PAC channel

Modulators often exhibit state–dependent binding to ion channels. This might arise from the fact that they can only access and exit their binding sites when the channels are in certain gating states (gated access mechanism) [26]. We first tested whether the inhibition of C77304 on the PAC channel specifically requires the channel to be in an open state, by using a sequential perfusion protocol as depicted in Fig 2A (upper panel). Specifically, PAC currents evoked by 2 consecutive pH 5.0 challenges, separated by a pH 7.3 incubation with or without saturating concentrations of C77304, were compared to evaluate any current inhibition at pH 7.3. It is worth noting that we performed a triple pH 7.3 bath solution replacement before the second pH 5.0 pulse to remove the residual compound in the chamber. Our data demonstrates that a saturating concentration (100 μM) of C77304 preincubated with wild–type (wt) PAC channels did not inhibit their currents (Fig 2A and 2B), suggesting that C77304 did not inhibit the wt–PAC channel in its resting state at pH 7.3. In contrast, the A321C mutant channel, which showed basal opening at pH 7.3 [9], was inhibited with this preincubation treatment (Fig 2A and 2B). Preincubation of the channels with bath solution (pH 7.3 only) did not affect their currents (Fig 2A and 2B).

Furthermore, the kinetics of preincubated C77304 inhibiting PAC currents also demonstrated that the inhibition was accompanied by and lagged channel activation, as assessed

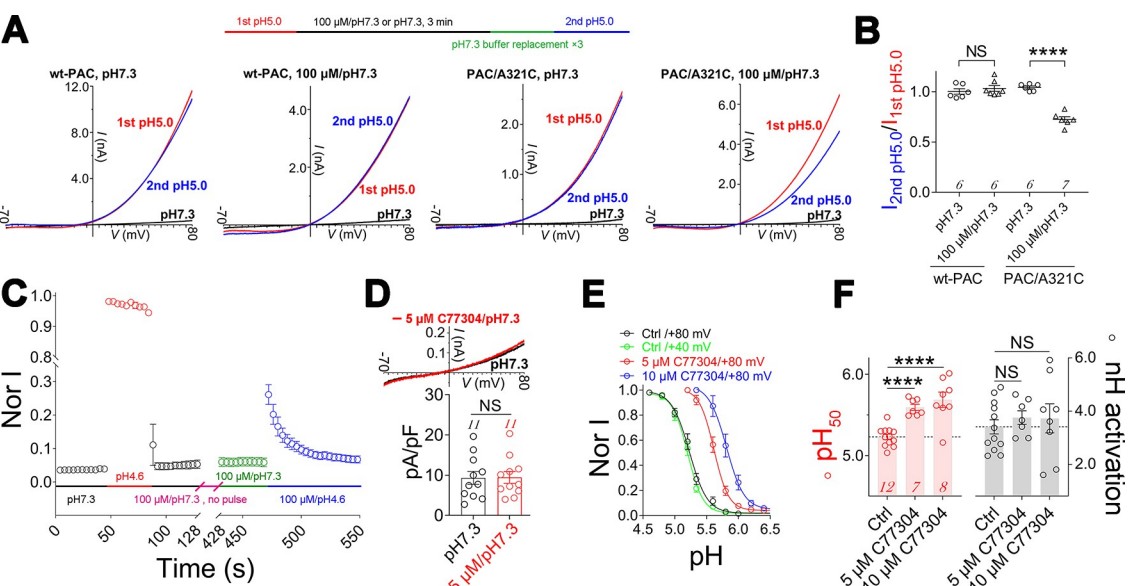

**Fig 2. The mechanism of action of C77304 on PAC channels.** (A) Upper panel: experimental protocol; lower 4 panels: representative traces showing 100 μM C77304 preincubated at pH 7.3 inhibits currents from A321C mutant, but not wt–PAC, channels ($n = 6$–7). (B) Summary of data in panel A ($n$ value as indicated in each group; NS, not significant; ****, $p < 0.0001$; unpaired $t$ test). (C) Time course of preincubated C77304 (100 μM/pH 7.3 for 5 min; see sequence of treatment conditions) blocking PAC currents upon channel activation by pH 4.6 (the blue circles) ($n = 9$). Currents were recorded at ramp depolarizations from −70 to +80 mV (1 mV/ms). (D) Example traces (upper panel) and summary analysis (lower panel) showing C77304 did not activate PAC currents at pH 7.3 ($n = 11$; NS, not significant; paired $t$ test). (E, F) Current–pH relationships of PAC channels before and after 5 and 10 μM C77304 treatment (E) and the related summary analysis of the $pH_{50}$s and slope factors (F), currents at +40 mV and/or +80 mV recorded by ramp depolarization (−70 to +80 mV) were analyzed ($n = 7$–12; NS, not significant; ****, $p < 0.0001$; one–way ANOVA with post hoc Dunnett analysis). The data underlying the graphs shown in the figure can be found in S1 Data. PAC, proton–activated chloride; wt, wild–type.

using the perfusion protocol in Fig 2C. The 5–min preincubation with 100 µM C77304 at pH 7.3 [magenta dashed bar in the perfusion sequence (100 µM/pH 7.3, no pulse), Fig 2C] was expected to fully inhibit PAC currents if the compound binds to the resting channels, thus a subsequent pH 4.6 test (blue bar in the perfusion sequence, Fig 2C) would detect no channel activity. As shown in Fig 2C, approximately 20% of PAC currents remained at the beginning of the second pH 4.6 pulse, which were then quickly inhibited as the channels became activated. Most (approximately 80%) channels might be immediately inhibited post activation, due to close proximity to C77304 and the fast inhibition kinetics. Notably, as stated in the preceding text, high concentrations of C77304 transiently activated PAC channels before inhibiting them in weak acidic conditions (Fig 1E), which is also in line with the idea that the inhibition requires the channel to be in an open state. Collectively, these data strongly argue that C77304 inhibits the PAC channel by binding to its open state.

C77304 did not potentiate PAC currents at pH 7.3 (Fig 2D), suggesting a proton gating–dependent activation. Indeed, C77304 concentration–dependently shifted the channel's current–pH relationship to the alkaline direction without changing the slope factor (Figs 2E, 2F and S2A and S1 Table), implying it increased the channel's apparent proton affinity by facilitating proton binding or enhancing proton gating, which were difficult to discriminate between. Additionally, proton binding to the PAC channel should be a voltage–independent event as the current–pH relationships measured at +40 mV and +80 mV are superimposed (Fig 2E). We further eliminated the possibility of C77304 inhibiting the PAC channel by driving its desensitization as the PAC/A321C mutant was directly inhibited at pH 7.3 but bifunctionally modulated at acidic pH 6.2 (S2B and S2C Fig). Moreover, the potentiating effect of C77304 was more pronounced at pHs at the onset ("root") of the current–pH curve (hereafter, "root pHs"), wherein most channels reside in the resting state (Figs 1G and 2E).

## The protonation–independent voltage gating of the PAC channel

PAC currents exhibit an outwardly rectifying behavior, wherein depolarization leads to an increase in the macroscopic conductance. This property can be explained by 2 mechanisms: voltage–dependent protonation and voltage gating that is independent of protonation. Our data supports the latter, since: (i) at pH 4.6, PAC channels were activated at very hyperpolarizing voltages and mediated small PS (pregnenolone sulfate [23], a PAC channel inhibitor)–sensitive inward currents (Fig 3A), suggesting the channels bind to protons voltage–independently and exhibit a small intrinsic opening probability when solely protonated; (ii) at pH 7.3, PAC channels were activated by strong ramp depolarizations from −195 mV to +195 mV, conducting large outwardly rectifying currents, which were absent in the PAC knock–out cells (Fig 3B–3D). Moreover, increasing pH from 7.3 to 8.3 and pH 9.3, respectively, only slightly attenuated the currents (Fig 3B and 3F), whereas reducing the proton concentration by 10–fold from pH 5.0 to 6.0, within the pH range where PAC channels were typically gated by protons, nearly eliminated the currents (Fig 3E and 3F), with the slope factor of the current–pH relationship by linear fit being determined as −0.22 and −0.95, respectively (Fig 3F). Notably, the current rundown caused by alkaline pH (pH 9.3) was quickly and fully reversed when the external pH recovered to pH 7.3 (S3A Fig). These data suggest that the PAC channel might be gated by alkali conditions as well, similar to that of the proton–gated proton channel OTOP1 [27]. As observed from the steady–state conductance–voltage relationship, the voltage–dependent activation of PAC channels at pH 7.3 illustrated a low voltage sensitivity and incomplete activation even under strong depolarization of +200 mV, with the maximal half activation voltage ($V_{1/2}$) and slope factor (K) being determined as 198.7 ± 13.4 mV and 31.7 ± 3.4 mV, respectively (S3B Fig).

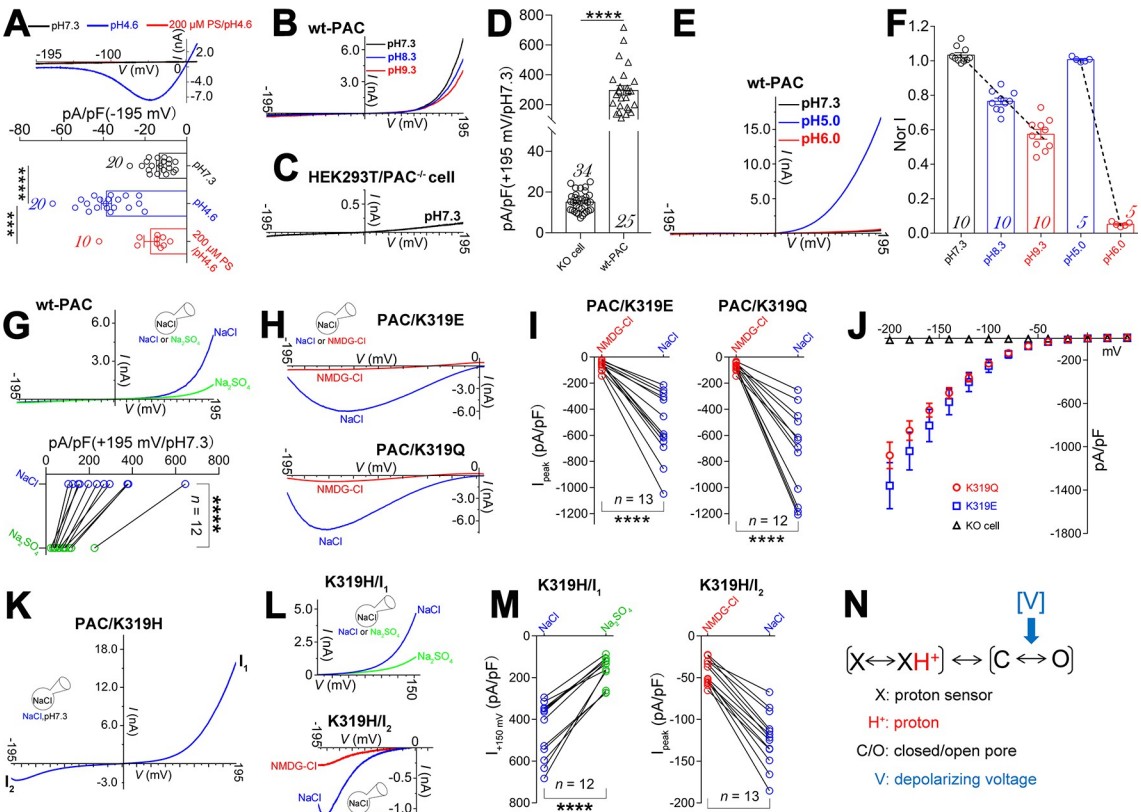

**Fig 3. Protonation–independent voltage gating in PAC channels.** (A) Representative traces (upper panel) and summary data (lower panel) showing pH 4.6 treatment activated a small but significant PS–sensitive and PAC channel–responsible currents at hyperpolarizing voltages. Currents were elicited by ramp depolarization from −195 mV to +10 mV ($n$ = 10–20; ****, $p < 0.0001$; ***, $p < 0.001$; paired $t$ test). (B–D) Representative current traces (B and C) and summary bar graphs (D) showing strong ramp depolarization (−195 mV to +195 mV) elicits large currents in wt–PAC transfected but not PAC knock–out cells at pH 7.3, with extracellular alkalization to pH 8.3 and 9.3 slightly reducing the amplitude (B) ($n$ = 10–34; ****, $p < 0.0001$; unpaired $t$ test). (E) Representative current traces showing pH 6.0 bath perfusion reduces the pH 5.0 acidification elicited currents in response to a −195 mV to +195 mV ramp depolarization ($n$ = 5). Untransfected HEK293T cells endogenously expressing the PAC channel were used for controlling the amplitude of acid–evoked currents at +195 mV. (F) Current–pH relationships of PAC channels in different pH ranges, with the linear fit slope factor determined to be −0.22 and −0.95 between pH 7.3–9.3 and pH 5.0–6.0, respectively; $n$ values indicated in each bar. Currents were recorded as in (B) and (E). (G) Representative traces (upper panel) and statistics (lower panel) showing $Na_2SO_4$ substitution of the external NaCl significantly reduces depolarization–activated PAC currents (****, $p < 0.0001$; paired $t$ test; $n$ = 12). (H, I) Representative current traces and statistics showing NMDG–Cl substitution of the external NaCl significantly reduces hyperpolarizaiton–activated currents in the PAC/K319E (H, upper panel; I, left panel) and PAC/K319Q (H, lower panel; I, right panel) channels (****, $p < 0.0001$; paired $t$ test; $n$ = 12–13). (J) I–V relationships of PAC/K319E and PAC/K319Q mutant channels at pH 7.3, currents were elicited by a cluster of voltage step from +40 mV to −200 mV (1 s) from the holding potential of 0 mV; HEK293T/PAC$^{−/−}$ (PAC knock–out) cell was included for comparison ($n$ = 10–15). (K) The PAC/K319H mutant channel conducted both inward and outward currents in response to a −195 to +195 mV ramp depolarization at pH 7.3 ($n$ = 7). (L, M) $Na_2SO_4$ and NMDG–Cl substitution of external NaCl significantly reduces the outward (L, upper panel; M, left panel) and inward currents (L, lower panel; M, right panel) in the PAC/K319H mutant channel, respectively (****, $p < 0.0001$; paired $t$ test; $n$ = 12–13). (N) Gating scheme of the PAC channel. The data underlying the graphs shown in the figure can be found in S1 Data. PAC, proton–activated chloride; PS, pregnenolone sulfate; wt, wild–type.

To gain a deeper understanding of the voltage gating mechanism in the PAC channel, our subsequent focus was on clarifying its voltage–sensing machinery. Introducing a mutation (K319E) at the lysine 319 position resulted in a conversion of the proton–activated PAC channel's selectivity from chloride ions (Cl$^−$) to sodium ions (Na$^+$) [11]. As anticipated, the wt PAC channel, which is strongly depolarization activated, allowed influx of chloride ions into the cell at pH 7.3. This was confirmed through an ion substitution assay, where external perfusion of

$Na_2SO_4$ resulted in a significant reduction of the outwardly rectified chloride currents (Fig 3G). However, the PAC/K319E mutant channel–mediated $Na^+$ influx into the cell at strong hyperpolarizing voltages at pH 7.3, as NMDG–Cl substitution of the external NaCl dramatically reduced the inward currents (Fig 3H, upper panel; Fig 3I, left panel; S3C Fig). The PAC/K319Q mutant channel phenocopied the PAC/K319E mutant channel (Fig 3H, lower panel; Fig 3I, right panel; S3D Fig). Notably, the PAC/K319E and PAC/K319Q mutants were likely activated very slowly, as their currents did not peak at the most hyperpolarizing voltage of −195 mV but at approximately −139.1 ± 2.5 mV (K319E) and −139.0 ± 2.2 mV (K319Q) (Figs 3H, S3C, and S3D), in response to a ramp depolarization from −195 mV to 0 mV, suggesting a time–dependent activation. Nevertheless, the PAC/K319E and PAC/K319Q mutant channels did conduct inwardly rectifying currents as assessed by testing their current–voltage relationships using step hyperpolarizing voltages (Figs 3J and S3E).

If we assume that the voltage sensor in the PAC channel consists of charged residues other than K319, we would expect that both the PAC/K319E and PAC/K319Q mutant channels, which retain the intact voltage sensing machinery like the wt PAC channel, would be activated by strong depolarization. However, this contradicts the experimental data presented in Fig 3H and 3J, as well as S3C–S3E Fig. Alternatively, if K319 is indeed part of the voltage sensor, a charge–reversal mutation like K319E could potentially reverse the voltage dependence of activation, causing the channel to be activated by hyperpolarizations. Conversely, neutralizing K319 (K319Q) would be expected to eliminate voltage gating. However, the experimental data provided in Figs 3H–3J, S3D, and S3E do not support this assumption either. Based on these findings, we can conclude that the PAC channel does not possess an inherent voltage sensor composed of the channel's own residues.

Instead, we found that the direction of current rectification in both the wild–type and mutant PAC channels is determined by the direction of selectively conducted ion flux. Specifically, $Cl^-$ influx driven by strong depolarizations through the wt–PAC channel, as well as $Na^+$ influx driven by strong hyperpolarizations through the PAC/K319E and PAC/K319Q mutant channels, increased the channels' open probability and resulted in outwardly rectified $Cl^-$ and inwardly rectified $Na^+$ currents, respectively (Figs 3G–3J and S3C–S3E). This conclusion was further validated in the PAC/K319H mutant channel. It allows both $Na^+$ influx at hyperpolarizing voltages and $Cl^-$ influx at depolarizing voltages at pH 7.3, mediating both inwardly rectified $Na^+$ and outwardly rectified $Cl^-$ currents, as $Na_2SO_4$ and NMDG–Cl substitution of the external NaCl significantly reduced the outward and inward currents, respectively (Fig 3K–3M). Altogether, these data support an ion–flux gating mechanism in the PAC channel, as reported for the K2P channel [28]. Consequently, a refined gating scheme for the PAC channel is depicted in Fig 3N, in which the voltage gating (V) and proton gating (X↔XH$^+$) work, both separately and synergistically, to drive pore opening (C↔O). These data implicate an intrinsic protonation–independent voltage gating machinery in the PAC channel. C77304 inhibited, but did not activate, the strong depolarization–activated PAC currents at pH 7.3 (S3F Fig), supporting its role in potentiating channel activation via actions on proton–, but not on voltage–gating. We propose that the conducting ions interacted with the selectivity filter residue (the 319th lysine, glutamate, glutamine, or histidine) to trigger voltage gating in the wild–type and mutant PAC channels.

## Mapping key residues involved in the proton gating of PAC channel and uncovering the putative binding sites of C77304

C77304 could be used as the pharmacological probe to identify the gating–related residues in PAC as it potentiated PAC currents only at root pHs. In this scenario, at pH 5.0, C77304 at

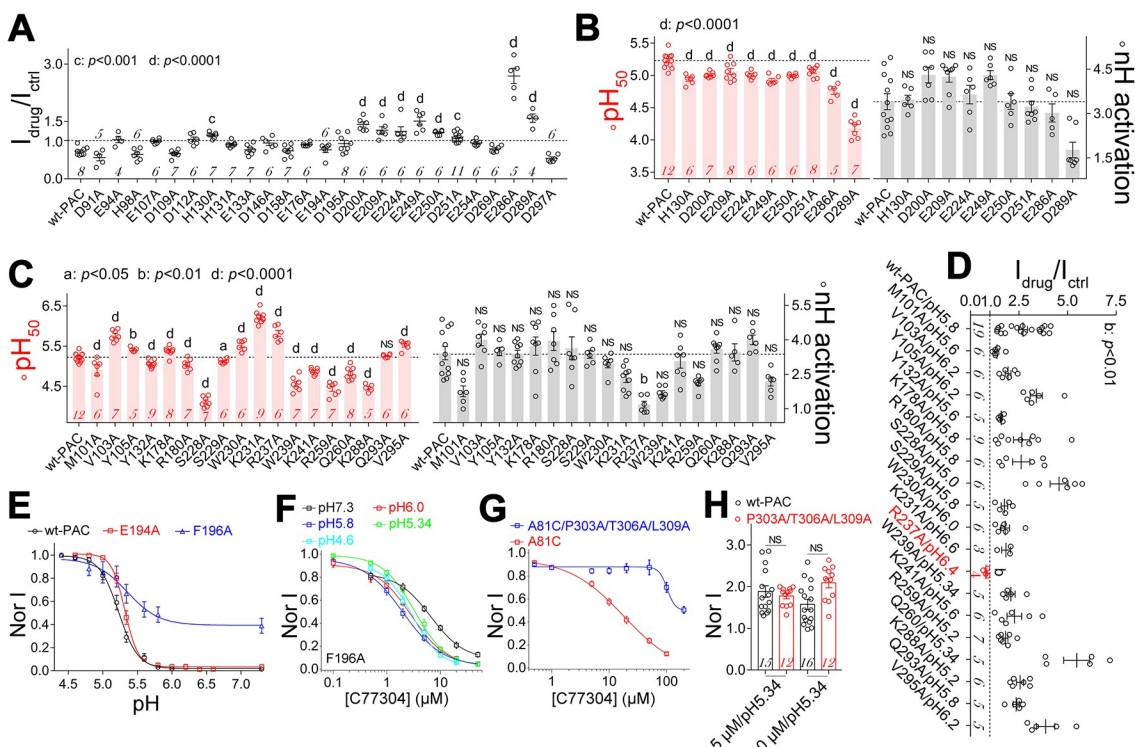

**Fig 4. Mapping key residues involved in proton gating and deciphering the putative C77304 binding sites in PAC channels.** (A) Alanine–scanning mutations of glutamate, aspartate, and histidine residues in the human PAC channel: Bar graphs show the normalized effect of 5 μM C77304 on their currents at pH 5.0. (B) Summary analysis showing the changes in $pH_{50}$ (left panel) and nH activation values (right panel) of PAC mutants activated by C77304 in (A). (C) Summary analysis showing most mutations in the side portal region of PAC channel significantly change the $pH_{50}$ of proton activation (left panel). The R237A mutation remarkably changed the slope factor (right panel). (D) Effects of 5 μM C77304 on PAC mutant channels' currents at their respective root pHs; currents were normalized to their respective control currents before drug treatment, showing the compound exclusively inhibited the PAC/R237A currents but activated the others. (E) Current–pH relationships of PAC/E194A and PAC/F196A mutant channels, wt–PAC was included for comparison ($n = 6$–$12$). (F) Dose–response relationships of C77304 inhibiting the PAC/F196A mutant channel at multiple pHs spanning its current–pH relationship. The $IC_{50}$s and mean slope factors were determined to be $6.6 \pm 0.9$ μM and 1.4 ($n = 8$), $2.9 \pm 0.2$ μM and 1.3 ($n = 6$), $2.0 \pm 0.2$ μM and 1.2 ($n = 7$), $3.6 \pm 0.3$ μM and 1.5 ($n = 5$), $3.0 \pm 0.5$ μM and 1.4 ($n = 7$), at pH 7.3, pH 6.0, pH 5.8, pH 5.34, and pH 4.6, respectively. (G) Concentration–response relationships of C77304 inhibiting the currents of A81C and A81C/P303A/T306A/P309A mutant channels at pH 7.3, with the $IC_{50}$s being determined as $18.3 \pm 3.0$ μM and approximately 200 μM, respectively ($n = 7$–$9$). (H) C77304 similarly activated the pH 5.34–evoked currents of the wt–PAC and P303A/T306A/P309A mutant channels; currents were normalized to that before drug treatment. In (A–D), the differences between mutant and wt–PAC channels were assessed by one–way ANOVA with post hoc Dunnett analysis; in (H), unpaired $t$ test was used; $p$ values as indicated in each panel and "NS" means not significant, $n$ values for each channel as indicated in the bar. The data underlying the graphs shown in the figure can be found in S1 Data. PAC, proton–activated chloride; wt, wild–type.

5 μM should activate, but not inhibit, those mutants with threshold activation pHs shifted to more acidic values. Indeed, among the 28 alanine–scan mutants of the titratable residues in PAC channel, the H130A, D200A, E209A, E224A, E249A, E250A, D251A, E286A, and D289A mutants were activated in this experimental setting (Figs 4A and S4A), suggesting an attenuation of their apparent proton sensitivity. Their current–pH relationships were shifted to the acidic direction to different extents, with remarkably reduced $pH_{50}$ but unchanged slope factor when compared to wt–PAC (Figs 4B and S4B and S1 Table), which reciprocally validated our hypothesis. Most importantly, we identified 3 glutamate residue mutations, E181A, E257A, and E261A, that fully abolished the proton gating of PAC (see Fig 5).

We proposed the existence of 2 C77304 binding sites on the PAC channel which underly its activation and inhibition effects. Indeed, these 2 sites were identified as the activation site 1

and the inhibition site 2 through subsequent mutation analysis (see below). Molecular docking of C77304 to the PAC channel (PDB: 7JNA [11]) revealed its preferred residence in the channel's side portal region. Consequently, we mutated residues comprising the side portal to alanines and measured their current–pH relationships, thus enabling determination of their respective root pHs for mapping the residues comprising the activation site 1. Most mutations profoundly affected channel gating: M101A, Y132A, R180A, S228A, S229A, W239A, K241, R259A, Q260A, and K288A mutations significantly reduced the channel's $pH_{50}$; V103A, Y105A, K178A, W230A, K231A, R237A, and V295A mutations increased $pH_{50}$ (Figs 4C and S4C and S1 Table). The R237A mutation also remarkably decreased the slope factor of the current–pH relationship (Fig 4C).

Mutating key residues in site 1 is expected to eliminate the activation effect of C77304. In fact, currents mediated by the PAC/R237A mutant, but not any others, were significantly inhibited when tested at their respective root pHs (Figs 4D and S4D). Furthermore, this inhibitory effect was also observed at multiple pHs tested (S4E Fig). Subsequent docking of C77304 to the channel around R237 predicted 4 additional candidate binding residues—E194, F196, K231, and W239 (S4F Fig). Among them, K231 and W239 mutations did not perturb the activating effect of C77304 (Fig 4D), suggesting that these were not key for anchoring C77304 in the pocket. The F196A mutation induced a large basal opening (approximately 39%) of the channel at neutral pHs (Fig 4E, S1 Table); moreover, the F196A but not the E194A mutation prevented activation by C77304, at their respective root pHs (S4E Fig). The concentration–response relationship of C77304 against the F196A mutant further confirmed that the compound monotonically inhibits its currents at multiple pHs spanning the current–pH relationship curve (Fig 4F). Combining the F196A and R237A mutations induced a smaller basal opening (approximately 14.8%) of the channel than the F196A mutation alone (S4G Fig). As expected, combining these 2 mutations also abolished C77304 activation (S4H and S4I Fig). In summary, these data show that F196 and R237 likely form activation site 1 in the PAC channel.

C77304 could inhibit both the proton–activated and voltage–activated PAC currents (Figs 1 and S3F), suggesting the compound acted on a common downstream step in these 2 gating processes. As deduced from the gating scheme illustrated in Fig 3N, the compound is most likely to inhibit the PAC channel by acting on the pore opening step. To identify key residues defining the inhibitory site 2, we used an alanine/cysteine–scan strategy to mutate residues in the pore–lining TM2 segment of the channel. An A81C mutation was introduced into these mutant channels to enable opening at pH 7.3 and then evaluated the inhibitory effect of C77304 without contamination from its activation effect. The P303A, T306A, and L309A mutations, but not the others tested, significantly attenuated the inhibitory effect of 20 μM C77304 (S5A and S5B Fig). Combining these mutations demonstrated an added effect (S5A and S5B Fig), which was further confirmed by comparing the concentration–dependent inhibition of C77304 on the A81C and A81C/P303A/T306A/L309A mutant channels (Fig 4G). Molecular docking experiments supported these findings (S5C Fig). Most importantly, the P303A/T306A/L309A mutations did not affect the activation effect of C77304 on the channel (Figs 4H and S5D). These data confirmed that distinct activation (site 1) and inhibition (site 2) sites coexist on the PAC channel.

Mapping the gating–related residues to PAC structure revealed their condensed distribution in the middle ECD, surrounding E181, E257, and E261 (S5E Fig). Most of these residues are not titratable at physiological pH and thus are unlikely the proton sensors. Instead, they might subtly adjust the biochemical environments where the proton sensors reside in to set their $pK_a$ or affect the gating of protonated channels.

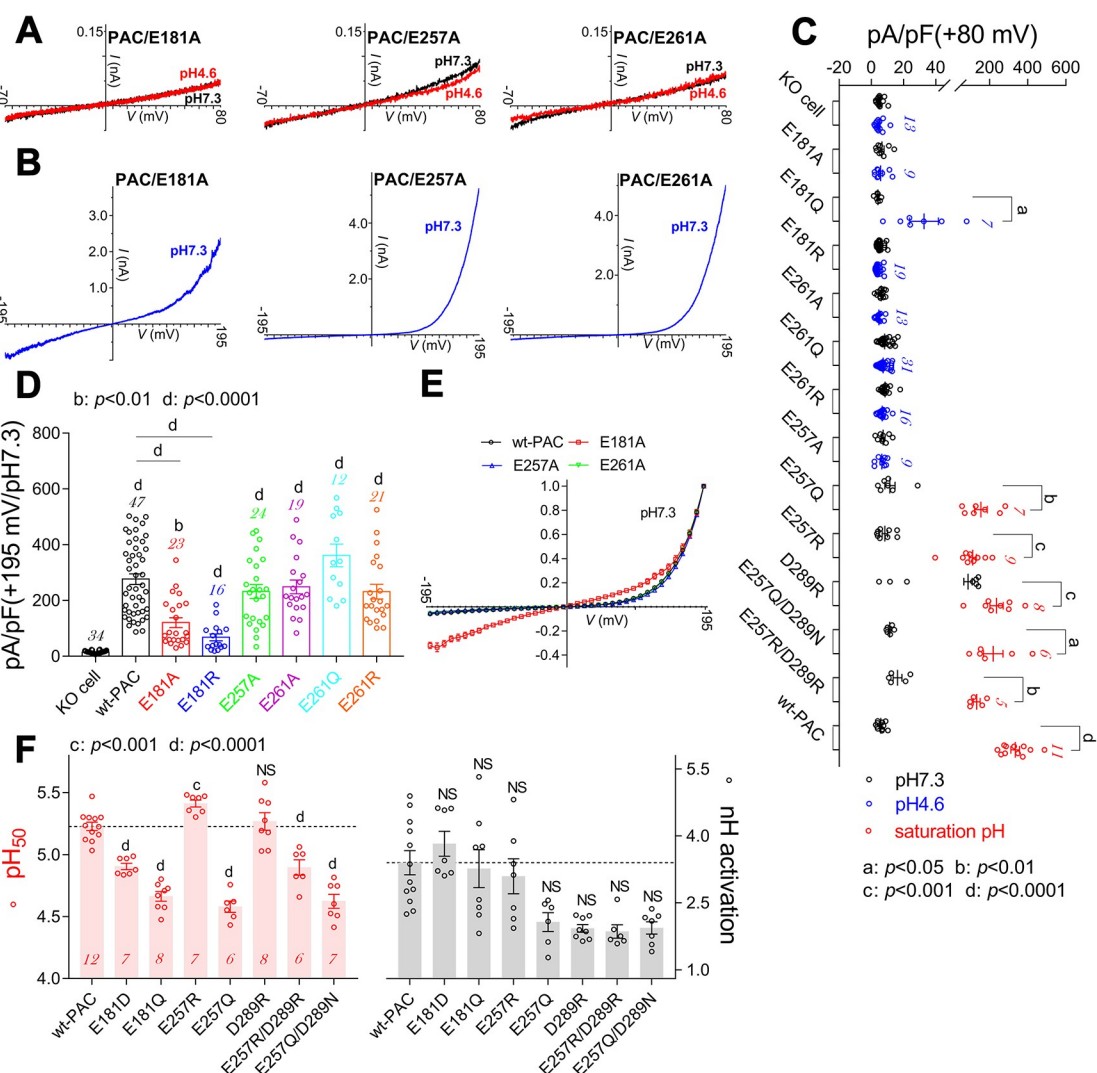

**Fig 5. Characterizing the primary proton sensors in human PAC channels.** (A, B) Representative current traces showing PAC/E181A, PAC/E257A, and PAC/E261A mutant channels did not respond to low pH$_o$ stimulation (A) but were activated by strong ramp depolarizations from −195 mV to +195 mV at pH 7.3 (B) ($n$ = 9–24). (C, D) Summary analysis of proton (C) and strong depolarization (D; at pH 7.3)–activated currents of mutant channels as indicated, the KO cell and wt–PAC groups were included for comparison ($n$ value as indicated in each bar). (E) Normalized I–V relationships of wt-PAC, PAC/E181A, PAC/E257A, and PAC/E261A mutant channels, with E181A mutation changed the I–V shape ($n$ = 8–12). (F) Summary analysis of pH$_{50}$ (left panel) and nH activation (right panels) values for mutant channels as indicated ($n$ values inside each bar). Statistical differences were assessed using paired $t$ test (C); one–way ANOVA with post hoc Dunnett analysis in (D) and (F); in (D), mutants were compared with KO and wt–PAC respectively; $p$ values as indicated in each panel and "NS" means not significant. Note in (D), the "NS" symbols between E257A, E261A, E261Q, E261R, and wt–PAC were omitted for clarity. The data underlying the graphs shown in the figure can be found in S1 Data. PAC, proton–activated chloride; wt, wild–type.

## Deciphering the primary proton sensors in the PAC channel

Alanine mutation of key proton sensing residues in the PAC channel should abolish the channel's proton activation by depleting its proton binding capability. Indeed, the E181A, E257A, and E261A mutations eliminated the channel's proton activation (Fig 5A and 5C). Immunocytochemistry experiments revealed that these mutant channels were targeted to cell membranes much like the wt–PAC channel (S6A Fig). Intriguingly, their voltage–dependent activation at pH 7.3 was not impaired, with peak current density for both the E257A and E261A mutant

channels being similar to the wt–PAC channel (Fig 5B and 5D). Moreover, the I–V relationships of the E257A and E261A mutant channels mirrored that of the wt–PAC channel, suggesting unchanged voltage sensitivity (Fig 5E). The E181A mutation, however, significantly altered the channel's I–V relationship and reduced the voltage–activated currents, implying its involvement in the channel's trafficking and/or voltage–dependent gating (Fig 5D and 5E).

Residue E257 was proposed to be involved in the proton gating of human PAC channel via its interaction with D289 from the neighboring subunit, as revealed by the open–state PAC channel structure [13]. Thus, we examined the potential role of the E257–D289 pairing in the channel's proton gating using a multiple–attribute mutation strategy (mutating E257 and D289 to R, Q, or N). All these mutants were functionally gated by protons (Figs 5C and S6B). Consistent with a previous study [13], the E257Q mutation strongly shifted the current–pH relationship to the acidic direction (Figs 5F and S6C and S1 Table). We reasoned that stabilizing the E257–D289 interaction would facilitate, while destroying it would impede channel gating. Indeed, the E257R mutation significantly shifted the current–pH relationship to the alkaline direction, whereas the D289R mutation produced a large basal opening at pH 7.3 without a $pH_{50}$ shift (Figs 5F and S6C and S1 Table). In contrast, the E257Q/D289N as well as the E257R/D289R double mutations caused a remarkable acidic–direction $pH_{50}$ shift (Figs 5F and S6C and S1 Table). If we assume that the E257–D289 pairing is fundamental for the proton activation of the PAC channel, then the E257R and D289R double mutations depleting this coupling should be able to eliminate the channel's proton gating; however, this was not observed experimentally. Therefore, these data argue that the E257–D289 interaction is not the primary driving force for the PAC channel's proton activation, but may facilitate channel gating.

Like E261A, the E261Q and E261R mutant channels were not functionally gated by protons (Figs 5C and S6D), but their voltage–dependent gating was not impaired (Figs 5D, S6E, and S6F). The multiple–attribute mutations of E181 exhibited an intermediate phenotype compared to the E257 and E261 mutations: The E181Q mutant channel was normally gated by protons but with significantly reduced $pH_{50}$ and current density (Figs 5C, 5F, S6G and S6I; S1 Table); the E181R mutant channel was refractory to proton gating with its voltage–dependent gating largely unaffected (Figs 5C, 5D and S6H). Surprisingly, the E181D mutation also caused a remarkable acidic–direction $pH_{50}$ shift (Figs 5F and S6I, S1 Table), suggesting the side chain length might affect the channel's proton gating as well. Altogether, these data implied that the E181, E257, and E261 mutations in the PAC channel eliminated the channel's proton gating mostly likely by depleting the proton binding, without affecting channel structure. These 3 residues should contribute to the primary proton sensors in the human PAC channel, although their individual role in channel gating might vary.

## Conservation and variation of the proton–sensing mechanisms among orthologous PAC channels

The PAC channel is an ancient ion channel broadly found across species in the animal kingdom, suggesting its conserved role throughout evolution. To further confirm the importance of the proton sensors identified in the human PAC channel, we performed cross–species analysis by mutating their analogous sites in various orthologous PAC channels from naked mole rat *H. glaber* (hgl–PAC), eastern brown snake *P. textilis* (pte–PAC), the Indian cobra *N. naja* (nna–PAC), red junglefowl *G. gallus* (gga–PAC), and zebrafish *D. rerio* (dre–PAC). As expected, all of these PAC channels functionally conducted proton–evoked (left panels in Figs 6A–6E and S7A, S7D, S7G, S7J and S7M) as well as voltage–activated (at pH 7.3) currents (right panels in Figs 6A–6E, S7A, S7D, S7G, S7J and S7M).

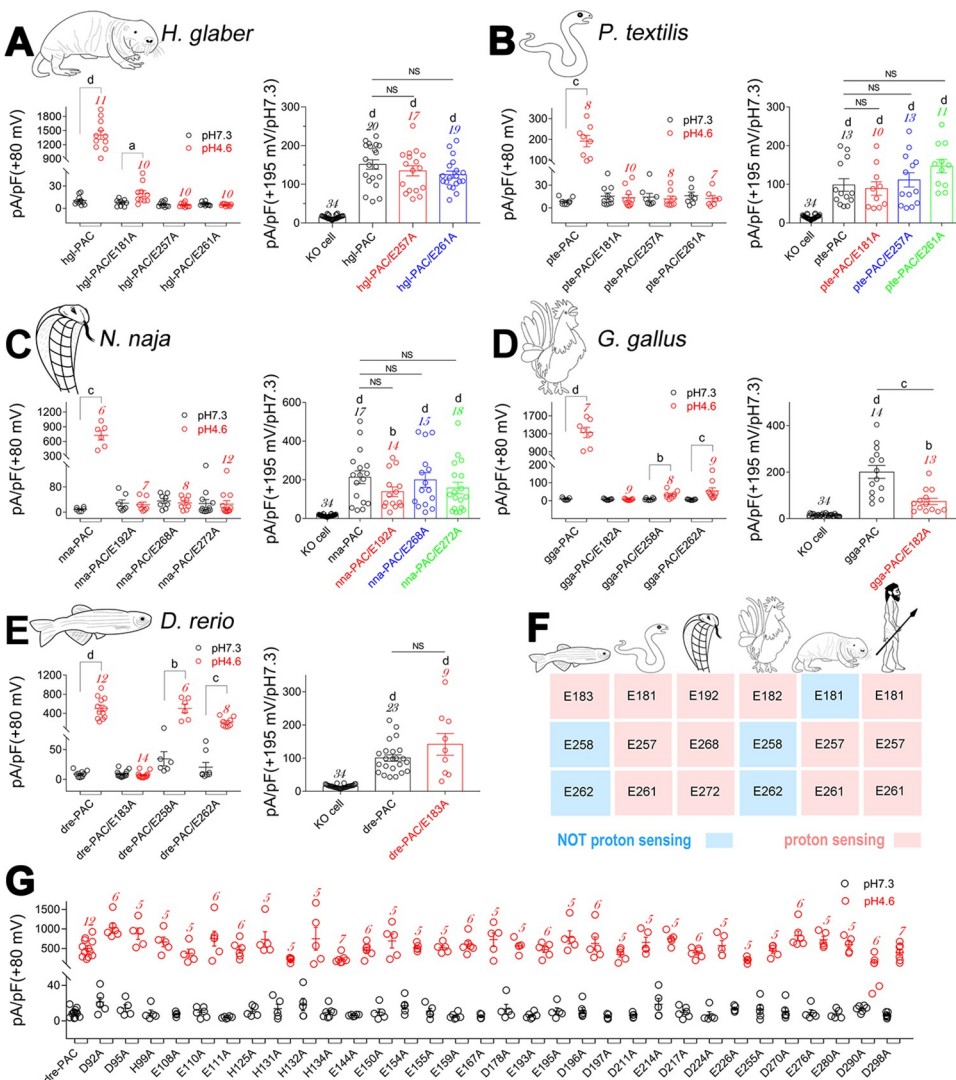

**Fig 6. Cross–species analysis of the proton sensing mechanism of PAC channels.** (A) Summary analysis of the proton (left) and strong–depolarization (right) activated currents of hgl–PAC, hgl–PAC/E181A, hgl–PAC/E257A, and hgl–PAC/E261A mutant channels from *H. glaber*. (B) Summary analysis of proton (left) and strong depolarization (right)–activated currents in pte–PAC, pte–PAC/E181A, pte–PAC/E257A, and pte–PAC/E261A mutant channels from *P. textilis*. (C) Summary analysis of proton (left) and strong depolarization (right)–activated currents of nna–PAC, nna–PAC/E192A, nna–PAC/E268A, and nna–PAC/E272A mutant channels from *N. naja*. (D) Summary analysis of proton (left) and strong depolarization (right)–activated currents of gga–PAC, gga–PAC/E182A, gga–PAC/E258A, and gga–PAC/E262A mutant channels from *G. gallus*. (E) Summary analysis of proton (left) and strong depolarization (right)–activated currents in dre–PAC, dre–PAC/E183A, dre–PAC/E258A, and dre–PAC/E262A mutant channels from *D. rerio*. (F) Heat map showing the homology of residues constituting the presumptive proton sensors among orthologous PAC channels. (G) Alanine–scan mutation of titratable residues in dre–PAC channel; all these mutants were effectively activated by protons by pH dropping from 7.3 to 4.6. Currents were recorded with −70 mV to +80 mV ramp depolarizations. Statistical differences between pH 7.3 and pH 4.6 in (A–E) were assessed by paired *t* tests; differences for strong depolarization–activated currents between groups in (A–E) were assessed by one–way ANOVA with post hoc Dunnett analysis (the KO cell and wt–PAC groups were used as control for comparison, respectively); a, $p < 0.05$; b, $p < 0.01$; c, $p < 0.001$; d, $p < 0.0001$; NS, not significant; *n* value as indicated in each bar. The data underlying the graphs shown in the figure can be found in S1 Data. PAC, proton–activated chloride; wt, wild–type.

Interestingly, mutating the E257 and E261 analogous sites in hgl–PAC, pte–PAC, and nna–PAC channels abolished their proton gating without affecting their voltage–dependent gating (Figs 6A–6C, S7B, S7C, S7E, S7F, S7H and S7I), which confirmed the critical role of these 2 residues in proton sensing of PAC channels across species. The E181 analogous site mutation in pte–PAC and nna–PAC channels also specifically eliminated their proton gating (Figs 6B, 6C, S7E, S7F, S7H and S7I). Surprisingly, the E181 analogous site mutation in the hgl–PAC channel did not fundamentally affect its proton gating, but did reduce the current density (left panel in Figs 6A and S7B).

Additionally, mutating the E181, but not E257 and E261, analogous residues in gga–PAC (E182A) and dre–PAC (E183A) channels specifically abolished their proton gating (Figs 6D, 6E, S7K, S7L, 7N and S7O). These data highlight conservation and variation in the proton–sensing machinery of orthologous PAC channels (Fig 6F). An extensive scanning–mutation analysis of all titratable residues in the dre–PAC channel also confirmed that the E183A mutation exclusively eliminated the channel's proton gating (Fig 6G). Consequently, the variation of proton sensors in orthologous PAC channels is likely limited to residues that are analogous to the 3 key proton–sensing residues identified in the human PAC channel—E181, E257, and E261(Fig 6F), with 1 or 2 key residues being abandoned during evolution in a species–specific manner.

## A gating model of PAC channel as revealed by MD simulations

MD simulations show that the ECD of PAC channel in the resting state contains a complex network of interactions within subunits (Fig 7A, left panel). For example, E261 interacts stably with K288 from the same subunit (Fig 7C and 7F). In the open state, the electrostatic repulsion between subunits is decreased, partially due to protonation of E261 (and probably others) at pH 4, leading to a significant decrease in intra–subunit interactions and increase in inter–subunit interactions (Fig 7A, right panel). Consequently, a less expanded ECD was observed (Fig 7B), consistent with its cryo–electron microscopic structure [11].

The neutralized E261 hydrogen bonds with Q260 from the adjacent subunit (Fig 7D and 7G). Notably, the E261–Q260 interactions are located at the neck of the ECD cavity, where its contraction begins in the active state and is essential for conformational transition. Simulations show that, in addition to E261 interacting with Q260 only (Model A in S8A Fig), E261 can also interact with W230 in the neighboring subunit (Model B in S8A Fig), thus interfering with the E261–Q260 link by pushing E261 outward (S8B Fig), rendering difficult the conformational transition of ECD upon protonation. In contrast, the W230A mutation eliminates this interference; these data are congruent with the W230A mutant being more sensitive to acidic environments (Fig 4C).

We further found that this interaction cannot be maintained in the E261Q mutant, and therefore, there was no ECD contraction (S8C Fig). This observation is consistent with experiment showing that the E261A/E261Q mutants lack acidic responses (Fig 5C). Presumably, the conformational rearrangement resulted from the altered protonation state of E261 is an integral part of its acid–sensitive mechanism, whereas the non–titrating residues are unable to play such a role.

E181 in the β6 strand of PAC channel was another crucial residue. MD simulations show that the side–chain of E181 stably forms a two–dentate interaction with the backbone of E176 and R175 on β5 that are stable in both the activated and resting states (Fig 7E and 7H). These interactions are essential for maintaining the coupling between β5 and β6 and are important links in the transmission of conformational changes. The E181D mutant with a shorter side–chain should weaken this coupling, making it less sensitive to acid. In addition, the charge–reversal E181R mutant was supposed to result in a complete loss of interaction. Both scenarios were proven correct in our experiments (Fig 5C and 5F).

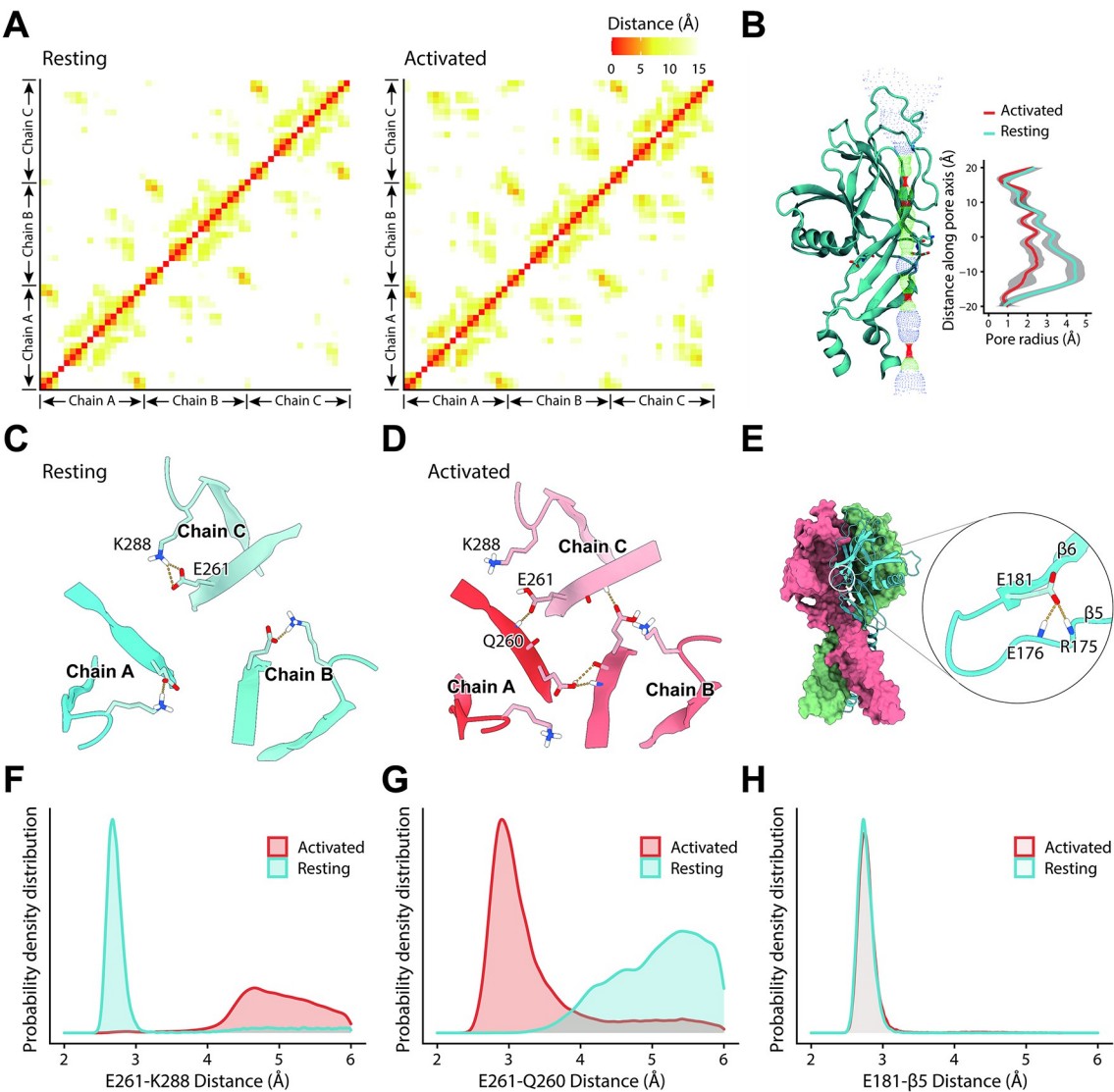

**Fig 7. A gating model of the PAC channel.** (A) Inter–and intra–chain contact maps in the resting state (left panel) and activated state (right panel). The x/y axis represents the residue ID on the 3 chains. The color bar indicates the distance (in angstroms) of the interactions. (B) The pore sizes of the PAC channel in the resting and activated states. Only 1 of the 3 subunits is shown for clarity. (C) The E261 interaction pattern in a representative resting state structure, dominated by electrostatic attractions within the subunit. The colors indicate the 3 subunits. (D) The E261 interaction pattern in a representative activated state structure, dominated by the hydrogen bond between Q260 and E261 on a neighboring chain in the activated state. (E) The interaction network involving E181 in a representative structure in the resting state taken from the MD trajectory. (F) The probability density distributions of the salt bridge distance between the carboxyl oxygen on E261 and the amino nitrogen on K288. (G) The probability density distributions of the distance between the carboxyl oxygen on E261's sidechain and the carboxyl oxygen or amide nitrogen on Q260's backbone. (H) The probability density distributions of the hydrogen bond distance between the carboxyl oxygen on E181's sidechain and the amide nitrogen on E176's or R175's backbone. The data underlying the graphs shown in the figure can be found in S1 Data. PAC, proton–activated chloride.

In our proposed gating mechanism, the ECD experiences protonation and tightens inter–subunit packing. As shown in S8D Fig, the conformation changes of ECD are translated to the TMD region, causing an outward rotation of the TMD segment. As mentioned previously, multiple amino acids (e.g., E261 and E181) in the ECD are involved in this process.

## Discussion

C77304 is a novel and unique bifunctional modulator of the PAC channel. That a modulator bifunctionally regulates ion channel activity is not unprecedented, as observed in the modulations of NaV1.3 channel by PF–06526290 and NaChBac channel by the general anesthetic drug sevoflurane [29,30]. One common feature of these promiscuous compounds is that they bind to different interacting sites on the target channel state–dependently. We have identified the putative C77304 binding sites on the PAC channel, designated the activation site 1 and inhibition site 2, respectively. The proton–binding site on the channel is denoted as site 0. Consequently, there are 8 permutations of their occupancy states (S2 Table). The binding of proton and/or C77304 to the PAC channel drives it state transitions, in which 16 different channel states theoretically exist depending on whether the channel's pore is open or closed (S2 Table). We simplified the model to contain 8 practically achievable states based on several experimentally determined constraints as addressed in Fig 2 and S2 Table. The model in S9A Fig fully recapitulates the bifunctional modulation of C77304 on the PAC channel, with C77304 binding to sites 1 and 2 with differing affinities (Fig 1G, 1I and 1J) and state–dependences (Fig 2) to regulate channel gating. Although C77304 binds to site 2 exclusively in PAC channel's open state (likely promote the $O_1 \rightarrow C_4$ and $O_2 \rightarrow C_5$ transitions; S9A Fig), it may activate the channel by binding to either the closed state to promote proton binding ($C_2 \rightarrow C_3 \rightarrow O_2$ and $C_2 \rightarrow O_2$ transitions; S9A Fig), the proton–bound pre–open state to enhance the pore opening ($C_1 \rightarrow O_2$ and $C_1 \rightarrow C_3 \rightarrow O_2$ transitions; S9A Fig), or even the open state to stabilize channel opening ($O_1 \rightarrow O_2$ transition; S9A Fig). We were unable to discriminate the indistinguishable binding effect and gating effect [31], both of which could contribute to the facilitated proton gating of the PAC channel by C77304. Accordingly, we incorporated all these mechanisms to build this state transition model. Importantly, C77304 can detect $pH_{50}$ shifts as small as approximately 0.15 pH units (Fig 4A and 4B) and is thus a valuable pharmacological probe for further dissecting the proton gating in PAC channel.

Polymodal gating is commonly observed in ion channels: TRPV1, TRPM8, TRPA1 channels in the transient receptor potential channel superfamily, the calcium–activated chloride channel TMEM16A, and the large–conductance calcium–activated potassium channel (BK), are gated by more than 1 stimuli such as chemicals, acid, $Ca^{2+}$, voltage, temperature, and mechanical force [32–36]. Membrane depolarization promotes the PAC channel opening in acidic conditions, resulting in outwardly rectified currents; however, whether the transmembrane voltage directly gates the PAC channel was unknown. Here, we show that the PAC channel is also activated by strong depolarizations even at alkaline pHs, which positions it as polymodally gated.

As for the voltage–dependent gating of PAC channel, 4 theoretically possible scenarios exist: (i) voltage and proton individually and allosterically activate PAC channel, resembling the voltage–and $Ca^{2+}$–dual–allosteric gating of the BK channel [37–39]; (ii) allosteric modulation that utilizes the proton gating pathway, as in the TRPV1 channel, where voltage gating involves the proton gating machinery [40], accordingly, protons cannot activate the channel at very negative hyperpolarizations; (iii) voltage activates PAC channel by potentiating its proton binding, like in the TMEM16A channel [41]; and (iv) voltage gates the channel through the conducting ions, that is the ion–flux gating mechanism as in the K2P channel [28]. Structural analysis revealed no charged residues in the transmembrane electrical field outside the channel pore, which eliminated the possibility of pore control by a remote voltage sensor and thus scenario 1. The experimental data showed that protons effectively activated PAC channels even at very negative hyperpolarizations (Fig 3A), which rules out scenario 2. Scenario 3 is invalidated

as the pH–current relationships of the PAC channel in acidic and neutral to alkaline pH ranges showed dramatically different slope factors (Fig 3F), suggesting distinct gating mechanisms. Most importantly, through an extensive mutation analysis of the selectivity filter (K319), we have confirmed that the PAC channel does not possess an endogenous built–in voltage sensor made up of channel residues (Fig 3); instead, we found that the direction of current rectification, which equates to a whole–cell conductance increase, is determined by the direction of the selectively conducted ion flux (Fig 3). These data strongly support scenario 4, that is, an ion–flux gating mechanism.

In physiological pH conditions, the PAC channel has a very high threshold activation voltage of around +100 mV. However, when exposed to protons, the threshold voltage is significantly shifted to approximately 0 mV and depolarizing voltages also greatly potentiate PAC channel opening, resulting in an outwardly rectified currents. This highlights the significance of voltage gating and suggests an interplay between voltage and proton gating. Moreover, it is unknown whether heat can also facilitate voltage–dependent gating of PAC as a previous study showed that increasing temperature enhances the channel's proton gating [42]. The coupling between voltage and proton gating, as well as the detailed mechanisms of voltage gating, need further study. It is likely that the residue constituting the selectivity filter in the PAC channel, K319, acts as the interacting site of the conducting ion to transmit the voltage gating signals.

The proton sensors in PAC channel remain undefined. As aforementioned, the Long group proposed that the E257–D289, E249–E107, and E250–D297 pairing in acidic conditions confer proton sensing in the human PAC channel [13]. In another study, the Qiu group proposed that the H98, H130, H131, and D269 residues form the proton sensor, for which the H98R/H130R/H131R/D269A quadruple mutations greatly shifted the channel's pH–current relationships to the alkaline direction accompanied by change of the slope factor and basal opening at neutral pH [43]. However, alanine–scanning mutagenesis of these residues (except for E257) did not eliminate the channel's proton gating, which raises questions about their roles in proton sensing. Instead, we found that the E181A, E257A, and E261A mutations eliminated the channel's proton gating (Fig 5), arguing for their critical roles in proton sensing. This is partially supported by another study published during the review process of our study [44]. The loss of proton–activated currents could also be attributed to the disruption of channel gating rather than proton sensing without evidence showing the mutants could be functionally gated, for example, by other stimuli. The Qiu group assigned E257 and E261 as components of a "joint region" that acts to relay allosteric signals from the proton sensors [43]. Here, by revealing the voltage–dependent gating of PAC channel at neutral pHs, we showed that E181A, E257, and E261 mutations specifically eliminated the proton gating of PAC channel without affecting the voltage gating, therefore supporting the assertion that these residues are the primary proton sensors. Indeed, incorporating the E257A and E261A mutations into the PAC/A321C mutant also abolished the channel's proton gating without affecting the proton–independent basal opening (S9B–S9D Fig), which is further evidence of their critical roles in proton sensing. Moreover, disrupting the proposed E257–K288 interaction [43] by the E257R mutation did not fundamentally affect the channel's proton gating (Fig 5C), therefore raising doubt as to its role in relaying gating signals. Cross–species analysis also validated the proton–sensing machinery we identified in the human PAC channel. Finally, sequence alignments showed that the E181, E257, and E261, but not the other gating–related residues are conserved across functional PAC channels from 13 different species (S10 Fig). Collectively, the identified proton sensors and C77304 binding sites should be invaluable in inspiring future drug development targeting the PAC channel.

## Materials and methods

### Compound synthesis

C77304 (5–iodo–2–(2–methylfuran–3–carboxamido)benzoicacid) was one "hit" compound of the manual patch–clamp screening of PAC channel modulator from a chemical compound library (from Selleck Chemicals LLC, Catalog No. L3600) and was synthesized by Nafu Biotechnology (Shanghai Nafu Biotechnology Co., Shanghai, China). The purity and structure of the synthetic C77304 was validated by HPLC and NMR analysis (S1D–S1G Fig).

### Cell culture, plasmids, and transient transfection

The HEK293T/PAC$^{-/-}$cell line was made by deleting a 31 bp long sequence (5′–AGCAGGA-CAAGGAGACGGTCAGAGTCCAAGG–3′) in exon 2 of the PAC coding gene PACC1 using the CRISPR–Cas9 method (Ubigene Biosciences, Guangzhou, China), the correct double knock–out the PACC1 gene was confirmed by PCR analysis and Sanger sequencing. KO cells were cultured in Dulbecco's Modified Eagle Medium (DMEM) (Invitrogen; Thermo Fisher Scientific, Waltham, Massachusetts, United States of America) supplemented with 10% FBS and 1% PS (all from Gibco; Thermo Fisher Scientific, Waltham, Massachusetts, USA), maintained at 37˚C in an incubator with saturated humidity and 5% $CO_2$. PAC coding genes from different species [*Homo sapiens* (accession number: NM_018252.3), *Heterocephalus glaber* (accession number: XM_004853543.3), *Pseudonaja textilis* (accession number: XM_026695787.1), *Naja naja* (accession number: ENSNNAG00000004337), *Danio rerio* (accession number: NM_001291762.1), *Gallus gallus* (accession number: XM_419431.8), *Anolis carolinensis* (accession number: XM_003216011.3), *Latimeria chalumnae* (accession number: XM_006013291.2), *Clupea harengus* (accession number: XM_012826242.3), *Poecilia reticulata* (accession number: XM_008398862.2), *Oreochromis niloticus* (accession number: XM_003442853.5), *Nothobranchius furzeri* (accession number: XM_015969346.1), and *Boleophthalmus pectinirostris* (accession number: XM_020921795.1)] were synthesized by Genscript (Genscript Corp., Nanjing, China) and cloned into the mammalian expression vector pCMV–blank. Channel mutants were made by site–directed mutations as previously described by us [45]. All constructs were sequenced to confirm correct mutations were made. Plasmids for wild–type or mutant PAC channels was transiently transfected into HEK293T/PAC$^{-/-}$cells using Lipofectamine per the manufacturer's instructions (Invitrogen; Thermo Fisher Scientific, Waltham, Massachusetts, USA). The transfection amount for each construct was adjusted based on its functional expression level in cells, and the transfection amounts were kept the same when comparing the current density from different mutant channels. Six hours after transfection, cells were seeded onto poly L–lysine (PLL)–coated coverslips, and patch–clamp analysis was conducted 24 to 36 h post transfection.

### Electrophysiology

Whole–cell patch clamp recordings were performed in an EPC–10 USB platform (HEKA Elektronik, Lambrecht, Germany). An agar bridge was used in electrophysiological recordings. Recording pipettes with an access resistance of 2 to 3 MΩ after pipette solution filling were prepared from glass capillaries in a PC–10 puller (NARISHIGE, Tokyo, Japan) using a two–steps program. To minimize pipette capacitance, only the tip of the pipette was filled with pipette solution. Artificial capacitance effect was reduced by sequential fast and slow capacitance compensation using the computer–controlled circuit of the amplifier. To minimize the leak current contamination, only cells with seal resistance higher than a gigaohm (GΩ) resistance after break in were selected for further analysis. Series resistance was kept to less than 10 MΩ to

minimize voltage error, and 80% series resistance compensation was used with a speed value of 10 μs. Cells were held at 0 mV, and PAC currents were elicited by a voltage ramp from −70 mV to +80 mV with a speed value of 1 mV/ms (or −195 mV to +195 mV with a speed value of 0.5 mV/ms for recording the strong depolarization activated currents at pH 7.3). Cell capacitance was measured using the LockIn Extention method when analyzing the current density. Recording solutions were as previously reported [8]: the extracellular solution contains (in mM): 145 NaCl, 2 KCl, 2 $MgCl_2$, 1.5 $CaCl_2$, 10 HEPES, 10 glucose (300 mOsm/kg; pH7.3 with NaOH). Different acidic pH solutions were made of the same ionic composition without HEPES but with 5 mM $Na_3$–citrate as buffer and the pH was adjusted using citric acid. The standard pipette solution contains (in mM): 135 CsCl, 1 $MgCl_2$, 2 $CaCl_2$, 10 HEPES, 5 EGTA, 4 MgATP (280 to 290 mOsm/kg; pH7.2 with CsOH). For the ion substitution experiments conducted in Fig 3G–3I and 3K–3M, the extracellular NaCl was replaced with $Na_2SO_4$ or NMDG–Cl. All chemicals were purchased from Sigma–Aldrich (Sigma–Aldrich, Saint Louis, Missouri, USA). Drugs and solutions were applied by gravity perfusion. Briefly, the solution in the perfusion chamber was removed by a vacuum system connected tubing, and the substituting solutions were applied simultaneously by a perfusion pipette placed near the recording pipette. This operation was repeated for 2 times to ensure complete replacement of the original solution in the chamber in most cases. The concentration–response curves were fitted by a Hill logistic equation to estimate the potency ($IC_{50}$ or $EC_{50}$) of C77304. The pH–current relationships of wt–PAC and PAC mutants were fitted by the equation: $Y = Y_{(0)} + (Y_{steady} − Y_{(0)})/(1 + 10^{\wedge}((pH_{50}−X)*nH))$, in which $pH_{50}$ and nH represents the half–activation pH and the slope factor of the curve, respectively. For analyzing the G–V relationship of PAC channels at pH 7.3, channels were clamped at 0 mV, and currents were elicited by a cluster of 2 s depolarizations from −100 mV to +200 mV (in 20 mV increment). The conductance (G) was calculated using the equation $G = I/(V−V_{rev})$, where I, V, and $V_{rev}$ represents the current amplitude, the depolarizing voltage, and the reversal voltage (0 mV), respectively. As the conductance (G) does not reach the maximum at +200 mV, the maximum G ($G_{max}$) cannot be experimentally determined. Instead, it was calculated by fitting the G–V relationship of each recording cell using the Boltzmann equation: $(Y−Y_{min})/(Y_{max}−Y_{min}) = 1/(1 + exp[(V_{1/2} −V)/K)])$, in which $V_{1/2}$, K, $Y_{max}$, and $Y_{min}$ represent the half maximum activation voltage, slope factor, maximum response, and minimum response, respectively. The conductance at 0 mV was estimated by the averaged values at −20 mV and +20 mV. The normalized G ($G/G_{max}$) was plotted as a function of V and the resulting curve was fitted using the Boltzmann equation.

## Immunocytochemistry

HEK293T/PAC$^{−/−}$cells transfected with human wt–PAC, PAC/E181A, PAC/E257A, or PAC/E261A mutant channel (with enhanced green fluorescense protein (EGFP) fused to their C–termini) were seeded on PLL–coated coverslips, stained with DiI and DAPI, and fixed with 4% PFA (all from Sigma–Aldrich). The coverslips were mounted in nail polish and observed under IX83 inverted fluorescence microscope (Olympus corporation, Tokyo, Japan), fluorescence of cell membrane (red), channel protein (green), and the nucleus (blue) was sequentially elicited and merged using the cellSens Dimension software (version 2.3; Olympus corporation, Tokyo, Japan).

## Molecular docking

The molecular docking was performed as previously described [46,47]. Rosetta Ligand application from Rosetta program suite version 2019 was used to dock C77304 to PAC channel and Monte Carlo algorithm was employed for sampling [48]. The three–dimensional

conformation of C77304 was generated by Frog2 server [49]. The structure of PAC channel (PDB ID: 7JNA) was relaxed using RosettaRelax procedure and the model with lowest score was used for further docking.

Docking included 3 stages: In the first, low–resolution stage, C77304 was initially placed into the pocket formed by 2 neighbored subunits in the upper ECD [11]; "center of mass" of C77304 was constrained to move within a 10 Å diameter sphere, where it could move freely during the docking process. The second, high–resolution stage employed the Monte Carlo minimization protocol in which the ligand position and orientation were randomly perturbed by a small deviation (0.1 Å and 3˚); channel residue side chains were repacked using a rotamer library; the ligand position, orientation, and torsions and protein sidechain torsions were simultaneously optimized using Quasi–Newton minimization and the end result was accepted or rejected based on the Metropolis criterion. The third and final stage was a more stringent gradient–based minimization of the ligand position, orientation, and torsions and the channel torsion angles for both side chains and backbone.

A total of 30,000 models were generated by docking and the top 10 models with lowest binding energy were chosen as candidates for further analysis.

### System setup and molecular dynamics simulations

For the resting state, 2 cryo–EM structure of PAC resting state (PDB entry: 7SQG [13] and 7JNA [11]) were used. For the open state, 2 cryo–EM structures (PDB entry: 7JNC [11] and 7SQF [13]) were used. Protein was embedded in a fully hydrated bilayer phospholipid membrane composed of 1–palmitoyl–2–oleoyl–glycero–3–phosphocholine (POPC) using CHARMM–GUI [50,51]. The residues' protonation states for the resting state in pH 8.0 and the open state in pH 4.0 were determined using PROPKA3 [52,53]. In addition to adding a few ions to neutralize the system, a 150 mM concentration NaCl was added to mimic the physiological condition of salt concentration. This system contains a total of approximately 157,000 atoms, including approximately 39,000 water molecules.

For both systems, energy minimization was first used to remove bad contacts, where harmonic restraints were applied on the protein backbone atoms. The simulation system was then equilibrated, with gradually decreased harmonic position restraints applied to the heavy atoms of the protein backbone. After equilibration, the simulations were continued in the NPT ensemble at 1 atm pressure and 303 K for 400 ns. The first 50 ns of each trajectory were not used for analysis. We carried out 2 replicate MD simulations for each system and obtained consistent results. The MD simulations were carried out using the CUDA–accelerated NAMD program version 2.14 [54]. The Charmm36 force field parameters [55] and the TIP3P water model [56] were used. Periodic boundary conditions were applied and the particle mesh Ewald method was used to treat long–range electrostatic interactions [57]. HOLE [58] was used to calculate the distribution of pore radius. Morph server [59] was used to generate the interpolated structure between the resting and activated structure. Autodock Vina [60] was used for docking. VMD [61] and UCSF ChimeraX [62] were used to analyze MD trajectories and visualize the models.

### Data analysis

Data were presented as MEAN ± SEM, $n$ value was presented as the number of separate experimental cells. Data were analyzed using the softwares Igor (WaveMetrics, Oregon, USA), Excel 2010 (Microsoft Corporation, Redmond, Washington, USA), OriginPro 8 (Northampton, Massachusetts, USA), and Graphpad Prism (GraphPad Software, La Jolla, California, USA). Statistics were analyzed using paired $t$ test, unpaired $t$ test, or one–way ANOVA, comparisons

between groups were performed using post hoc analysis with the Dunnett method. Statistical difference was accepted at $p < 0.05$. The numerical data used in all figures are included in S1 Data.

## Supporting information

**S1 Data.** Excel spreadsheet containing, in separate sheets, the underlying numerical data and statistical analysis for Fig panels 1A (lower panel), 1E (lower panel), 1F (lower panel), 1G, 1H, 1I, 1J, 1K, 1L, 2B, 2C, 2D (lower panel), 2E, 2F, 3A (lower panel), 3D, 3F, 3G (lower panel), 3I, 3J, 3M, 4A, 4B, 4C, 4D, 4E, 4F, 4G, 4H, 5C, 5D, 5E, 5F, 6A, 6B, 6C, 6D, 6E, 6G, 7A, 7B (right panel), 7F, 7G, 7H, S2C, S3A (right panel), S3B (right panel), S3F (right panel), S4B, S4C, S4E, S4G, S4I, S5B, S6C, S6I, S8C, and S9D.
(XLSX)

**S1 Fig.** (A–C) Structures of 3 backup "hit" compounds (upper panels) and representative current traces showing their inhibition of pH 4.6– and pH 5.34–evoked PAC currents (lower panels). C65841, C62975, and C66386 (at 10 μM) inhibited the pH 4.6–evoked PAC currents by 64.8 ± 0.5%, 77.8 ± 2.9%, and 53.3 ± 1.5%, and the pH 5.34–evoked PAC currents by 35.4 ± 3.2%, 47.8 ± 1.9%, and 28.0 ± 2.6%, respectively ($n = 3$). Note that the background leak currents (currents at pH 7.3) were subtracted from the total currents when calculating the inhibition ratio at pH 5.34, considering the relatively small amplitude of the pH 5.34–evoked currents. (D, E) Liquid chromatography (LC) analysis demonstrated that the purity of synthesized C77304 is above 99% as determined by the peak area normalization method at (D), 214 nm and (E), 254 nm. (F, G) Structure validation of synthesized C77304 by $^1$H–NMR (F) and $^{13}$C–NMR (G). (H) Predicted ionization ratio ([A$^-$]/([HA] + [A$^-$]), denoted as "% ionized," in which [A$^-$] and [HA] represents the concentration of deprotonated and uncharged C77304 molecules, respectively) of C77304 at pHs ranging from 1.0 to 7.4, with the range of interest (pH 4.6 to 7.4) marked with pink box. The p$K_a$ of the carboxy group in C77304 was calculated to be 3.64 and 2.922 (mean values) by Epik and ChemDraw, respectively. The [A$^-$]/[HA] ratio at each pH was calculated using the Henderson–Hasselbalch equation: $pH = pK_a + \log([A^-]/[HA])$.
(TIF)

**S2 Fig.** (A) Representative PAC currents elicited by different acidic pH solutions in the absence (left) and presence of 5 μM C77304 (middle) or 10 μM C77304 (right). Note that the C77304 treatment changed the threshold and saturating activation pHs. Currents were recorded with ramp depolarizations from −70 mV to +80 mV ($n = 7$–12 cells per condition). (B) Representative current traces showing C77304 monotonically inhibits (left) or bidirectionally modulates (activating and inhibiting; right) PAC/A321C mutant channels at pH 7.3 and pH 6.2, respectively ($n = 5$). (C) Concentration–response relationships of C77304 acting on PAC/A321C mutant channel, with the curves being Sigmoidal or bell–shaped at pH 7.3 and pH 6.2, respectively. The EC$_{50}$ for C77304 activating PAC currents at pH 6.2 was 1.0 ± 0.4 μM, and the IC$_{50}$ for inhibition were 11.8 ± 1.9 μM and 20.0 ± 2.9 μM at pH 7.3 and pH 6.2, respectively ($n = 5$). The data underlying the graphs shown in the figure can be found in S1 Data.
(TIF)

**S3 Fig.** (A) Representative current traces (left) and summary of normalized currents (Nor I, right) showing alkali treatment reversibly reduced the strong depolarization–activated PAC currents ($n = 11$); statistics were evaluated by paired $t$ tests (****, $p < 0.0001$; NS, not significant). (B) Representative current traces of wt–PAC channels elicited by step depolarizations from −100 mV to +200 mV (in 20 mV increment) at pH 7.3 (left panel) and the normalized

conductance vs. voltage relationship (right panel). The half maximal activation voltage ($V_{1/2}$) and the slope factor (K) were determined as $198.7 \pm 13.4$ mV and $31.7 \pm 3.4$ mV, respectively ($n = 9$). (C, D) Representative current traces of PAC/K319E (C) and PAC/K319Q (D) mutant channels elicited by ramp depolarizations from −195 to +195 mV at pH 7.3 ($n = 7$). (E) Representative current traces of PAC/K319E and PAC/K319Q mutant channels activated by depolarizing/hyperpolarizing voltage steps from +40 to −200 mV (in increments of 20 mV) at pH 7.3. The HEK293T/PAC$^{-/-}$(KO) cell (lacking any PAC channels) was used as the control ($n = 10$–15). (F) Representative traces (left) and summary of normalized currents (right) showing that C77304, in a concentration–dependent manner, inhibited the strong depolarization–activated PAC currents at pH 7.3 ($n = 10$). The data underlying the graphs shown in the figure can be found in S1 Data.
(TIF)

**S4 Fig.** (A) Representative current traces showing the effect of 5 µM C77304 on PAC channel mutants of the titratable glutamate, aspartate, and histidine residues at pH 5.0; voltage protocol as shown ($n = 4$–11). (B) Current–pH relationships of the mutant PAC channels as indicated, showing differing extent of shifts to more acidic values compared with the wild–type (wt)–PAC ($n = 5$–12). (C) Current–pH relationships showing that mutating residues in the side portal region of the PAC channel affects its proton gating ($n = 5$–12). (D) Typical current traces demonstrating the effect of 5 µM C77304 on PAC channel mutants of residues in the side portal region at their respective threshold pHs; voltage protocol as shown ($n = 3$–9). (E) Effects of 5 µM C77304 on the currents of PAC/R237A (at various pHs), PAC/E194A (at threshold pH), and PAC/F196A (at threshold pH) mutant channels, showing that the F196A and R237A mutations effectively eliminated the compound's activating effect ($n$ values as indicated in each bar). (F) Molecular docking of C77304 to the PAC channel around the R237 site revealed that residues E194, F196, K231, R237, and W239 form the putative binding pocket. (G) Current–pH relationship of the PAC/F196A/R237A mutant channel, with the pH$_{50}$ and basal opening proportion (BOP) at neutral pHs being determined to be $5.47 \pm 0.1$ and $0.15 \pm 0.02$ ($n = 7$). (H, I) Representative current traces (H) and concentration–response relationships (I) of C77304 inhibition of the PAC/F196A/R237A mutant channel. The IC$_{50}$s were calculated to be $5.7 \pm 0.9$ µM and $3.5 \pm 0.9$ µM at pH 7.0 and pH 6.4, respectively ($n = 10$–12). The data underlying the graphs shown in the figure can be found in S1 Data.
(TIF)

**S5 Fig.** (A) Representative current traces showing reduced inhibition of 20 µM C77304 on PAC/A81C/P303A, PAC/A81C/T306A, PAC/A81C/L309A, PAC/A81C/P303A/T306A/L309A mutant channels when compared with the PAC/A81C mutant control ($n = 6$–12). (B) Summary analysis of the inhibitory effect of 20 µM C77304 on PAC channel mutants of residues in the pore–lining TM2 segment. Currents were recorded at ramp depolarizations from −70 to +80 mV at pH 7.3. The P303A, T306A, L309A, and P303A/T306A/L309A triple mutations, but not others, significantly attenuated the compound's inhibitory effect ($n$ value as indicated in each bar; the differences between mutant channels and the PAC/A81C mutant control were assessed by one–way ANOVA with post hoc Dunnett analysis). (C) The docking pose for C77304 in the inhibition pocket. Residue L309 stacks with the aromatic ring of C77304 through hydrophobic interactions. Residues P303 and T306 above the pocket may stabilize the conformation of the pocket by internal tension and hydrogen bonding, respectively. (D) Representative current traces showing C77304 apparently potentiated the currents of PAC/P303A/T306A/L309A mutant channel at pH 5.34 ($n = 12$). (E) PAC channel structure (PDB entry: 7SQG) showing locations of mutation sites as indicated. Bold text indicates mutations that had the strongest effects. At far right, domains and secondary structure elements are labeled

according to the report by Osei–Owusu and colleagues. The data underlying the graphs shown in the figure can be found in S1 Data.
(TIF)

**S6 Fig.** (A) Immunocytochemistry imaging revealed the membrane expression of GFP–tagged mutant channels (green fluorescence), co–localized with the membrane marker DiI (red fluorescence). Representative micrographs from 3 independent experiments are shown. (B) Representative current traces showing that the PAC/E257Q, PAC/E257R, PAC/D289R, PAC–E257Q/D289N, and PAC–E257R/D289R mutant channels are functionally gated by protons ($n = 5$–9). (C) Current–pH relationships of PAC mutant channels as indicated, showing E257R, E257Q, E257Q/D289N, and E257R/D289R mutations shifted the curve to either acidic or alkaline directions, and D289R mutation caused a considerable basal opening at pH 7.3 ($n = 6$–12). (D–F) Example current traces showing that the PAC/E261Q and PAC/E261R mutant channels were not functionally gated by proton (D) but by strong ramp depolarization at pH 7.3 (E and F) ($n = 12$–31). (G) Typical proton–activated current of PAC/E181Q mutant channel ($n = 7$). (H) PAC/E181R mutant channel in response to acid (upper panel) and strong ramp depolarization at pH 7.3 (lower panel) ($n = 16$–19). (I) Current–pH relationships of PAC/E181Q and PAC/E181D mutant channels compared with wt–PAC ($n = 7$–12). The data underlying the graphs shown in the figure can be found in S1 Data.
(TIF)

**S7 Fig. Cross–species analysis of the proton–sensing mechanism of PAC channels.** (A) Typical proton–or strong depolarization–activated currents of the PAC channel from *H. glaber* (hgl–PAC) ($n = 11$–20). (B) Mutating the human PAC E257 and E261 analogous sites (hgl–PAC/E257A and hgl–PAC/E261A), but not the E181 analogous site (hgl–PAC/E181A), in hgl–PAC eliminated its proton gating ($n = 10$–13). (C) Representative current traces showing hgl–PAC/E257A and hgl–PAC/E261A channels were activated by strong ramp depolarizations from −195 mV to +195 mV at pH 7.3 ($n = 17$–19). (D) Representative proton–or strong depolarization–activated currents of the PAC channel from *P. textilis* (pte–PAC channel) ($n = 8$–13). (E) Example current traces showing mutating all the 3 residues analogous to the human PAC proton–sensing sites in pte–PAC (pte–PAC/E181A, pte–PAC/E257A, and pte–PAC/E261A) fully eliminated its proton gating ($n = 7$–10). (F) Representative strong depolarization–activated currents of pte–PAC/E181A, pte–PAC/E257A, and pte–PAC/E261A mutant channels at pH 7.3 ($n = 10$–13). (G) Typical proton–or strong depolarization–evoked currents of the PAC channel from *N. naja* (nna–PAC channel) ($n = 6$–17). (H) Representative current traces showing E192A, E268A, and E272A mutations in nna–PAC (analogous to the human PAC E181, E257, and E261 site, respectively) abolished its proton gating ($n = 7$–12). (I) Typical strong depolarization–activated currents of nna–PAC/E192A, nna–PAC/E268A, and nna–PAC/E272A mutant channels at pH 7.3 ($n = 14$–18). (J) Typical proton–or strong depolarization–activated currents of the PAC channel from *G. gallus* (gga–PAC channel) ($n = 7$–14). (K) Example current traces showing E182A (analogous to human PAC E181) but not E258A and E262A (analogous to human PAC E257 and E261, respectively) mutations in gga–PAC eliminated its proton gating ($n = 8$–9). (L) Typical strong depolarization–activated currents of gga–PAC/E182A mutant channel at pH 7.3 ($n = 13$). (M) Typical proton–or strong depolarization–activated currents of the PAC channel from *D. rerio* (dre–PAC channel) ($n = 12$–23). (N) Representative current trace demonstrating that mutating residues analogous to human PAC E181 but not E257 and E261 in dre–PAC (dre–PAC/E183A but not dre–PAC/E258A and dre–PAC–E262A) fully eliminated its proton gating ($n = 6$–14). (O) Typical strong depolarization–activated currents of dre–PAC/E183A mutant channel at pH 7.3 ($n = 9$).
(TIF)

**S8 Fig. Gating model of the PAC channel.** (A) The interactions between Q260 and the E261–W230 cluster on a neighboring chain in the activated state. The different orientations of E261 lead to models A (left) and B (right). (B) The probability density distributions of the distance between the carboxyl oxygens on E261 and on Q260 in the 2 models. (C) The pore sizes of the wild–type PAC channel in the resting and activated states and the E261Q mutant PAC channel in the activated state. The pore sizes are sampled from the MD trajectory of the system containing only ECD. (D) The predicted structural transformation from resting state to activated state were made using Morph. The data underlying the graphs shown in the figure can be found in S1 Data.
(TIF)

**S9 Fig.** (A) Proposed model illustrating the binding/unbinding of protons and/or C77304 with PAC channel and the resulting state transitions. "$C_0$–$C_5$" and "$O_1$–$O_2$" represent different closed and open states, respectively. Site 0, proton binding; site 1, activation site; site 2, inhibition site. (B, C) Representative current traces demonstrating that the PAC–E257A/A321C (B) and PAC–E261A/A321C (C) double mutant channels are not activated by protons but exhibit considerable basal opening at pH 7.3 ($n = 12$). Currents were recorded with a voltage ramp from −70 mV to +80 mV. (D) Summary analysis of the proton–activated currents of PAC/A321C, PAC–E257A/A321C, and PAC–E261A/A321C mutant channels, significant differences between the pH 7.3 and pH 4.6 conditions were assessed using a paired $t$ test; $p$ value as indicated in the panel and $n$ values as indicated in each bar. The data underlying the graphs shown in the figure can be found in S1 Data.
(TIF)

**S10 Fig.** (A) Sequence alignment of PAC channels from 13 different species as indicated, showing that residues E181, E257, and E261, but not the other gating–related residues are conserved across species (in human PAC numbering). In PAC channels from *P. reticulata*, *O. niloticus*, and *N. furzeri*, the proposed proton sensor H98 is mutated to glutamine or asparagine; in PAC channel from *B. pectinirostris*, the proposed proton sensors H98 and H131 are mutated to asparagine and tyrosine, respectively; in PAC channels from *C. harengus*, *P. reticulata*, *O. niloticus*, *N. furzeri*, and *B. pectinirostris*, the E249 and D297 analogous sites mutation are expected to abolish the proposed E107–E249 and E250–D297 carboxy–carboxylate interactions in the activated state; the D109, E250, and Q296 residues which were proposed to form the H98 binding pocket are also varied among orthologous PAC channels from lots of species. (B) Typical proton–activated currents of PAC channels from *A. carolinensis*, *L. chalumnae*, *C. harengus*, *P. reticulata*, *O. niloticus*, *N. furzeri*, and *B. pectinirostris* ($n = 4$–9).
(TIF)

**S1 Table. Summary of proton gating parameters of wild–type (wt) and mutant PAC channels tested in this study.** Abbreviations: $pH_{50}$, half activation pH; nH activation, slope factor of pH–current relationship; BOP, basal opening proportion at neutral pHs as determined by fitting the pH–current relationships; $n$, number of separate experimental cells; a, $p < 0.05$; b, $p < 0.01$; c, $p < 0.001$; d, $p < 0.0001$; statistical differences were assessed by one–way ANOVA with post hoc analysis using the Dunnett method (wt–PAC as control). Data are presented as MEAN ± SEM.
(XLSX)

**S2 Table. Deduction of the PAC channel states upon proton and/or C77304 binding.** √: site occupied; ×: site unoccupied; C: pore closed; O: pore opening; $C_0$, $C_1$, $C_2$, $C_3$, $C_4$, and $C_5$ represent different closed states of the PAC channel, and $O_1$ and $O_2$ represent 2 open states.

The proton binding site on the PAC channel was defined as site 0, and the 2 C77304 binding sites on it were denoted as site 1 and site 2, which mediate the activation and inhibition effect of the compound, respectively. There are 8 possible combinations of the occupancy situations of site 0, site 1, and site 2 by proton and/or the compound (ranging from no site being occupied to all 3 sites being occupied). Taking into consideration the 2 pore states (open or closed), there exist theoretically 16 different channel states. Several experimentally determined constraints reduced the channel states: (i) C77304 binding to site 1 alone cannot drive the channel opening; (ii) C77304 binding to site 2 requires the channel in its open state, which means site 0 must be bound with protons if site 2 was to be occupied; (iii) the channel would be trapped in the closed state as long as site 2 is bound with C77304; and (iv) the pore of the PAC channel is highly unlikely to be opened without protons binding with site 0. These constraints excluded 8 possible channel states in this table (highlighted with shadows) and only 8 channel states are therefore practically achievable.

(XLSX)

## Author Contributions

**Conceptualization:** Cheng Tang.

**Data curation:** Cheng Tang, Fan Yang.

**Formal analysis:** Cheng Tang.

**Funding acquisition:** Cheng Tang, Zhonghua Liu.

**Investigation:** Piao Zhao, Cheng Tang, Yuqin Yang, Zhen Xiao, Samantha Perez-Miller, Heng Zhang, Guoqing Luo, Hao Liu, Yaqi Li, Qingyi Liao.

**Methodology:** Cheng Tang.

**Supervision:** Cheng Tang, Hao Dong, Rajesh Khanna, Zhonghua Liu.

**Writing – original draft:** Cheng Tang, Hao Dong, Rajesh Khanna.

**Writing – review & editing:** Cheng Tang, Hao Dong, Rajesh Khanna.

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
