## [Editor Report · Decision Letter 0]

31 Mar 2023

Dear Dr Tang, 

Thank you for submitting the revised version of your manuscript entitled "A new polymodal gating model of the proton-activated chloride channel" for consideration as a Research Article by PLOS Biology.

Your revision has now been evaluated by the PLOS Biology editorial staff as well as by the original academic editor and I am writing to let you know that we would like to send your submission back to the original reviewers.

However, because this is a new submission, we need you to provide again the metadata that is required. To this end, please login to Editorial Manager where you will find the paper in the 'Submissions Needing Revisions' folder on your homepage. Please click 'Revise Submission' from the Action Links and complete all additional questions in the submission questionnaire.

Once your full submission is complete, your paper will undergo a series of checks in preparation for peer review. After your manuscript has passed the checks it will be sent back to the reviewers. To provide the metadata for your submission, please Login to Editorial Manager (https://www.editorialmanager.com/pbiology) within two working days, i.e. by Apr 04 2023 11:59PM.

Kind regards,

Ines

--

Ines Alvarez-Garcia, PhD

Senior Editor

PLOS Biology

---

## [Decision Letter · Decision Letter 1]

2 Jun 2023

Dear Dr Tang,

Thank you for your patience while we considered your revised manuscript entitled "A new polymodal gating model of the proton-activated chloride channel" for publication as a Research Article at PLOS Biology. Your revised study has been evaluated by the PLOS Biology editors, the Academic Editor and two of the original reviewers.

The reviews are attached below. You will see that the two reviewers acknowledge the improvements you have done in the revision and that you have addressed some of the concerns that were previously raised. However, they also think that several points remain to be addressed. Reviewer 2 mentions that the mechanism proposed based on a kinetic model has not been experimentally validated, thus the reviewer suggests moving the model to the discussion. In addition, there is not discrimination between rectification in the IV curve as a consequence of an altered open probability (reflecting gating) or as a consequence of a non-uniform conductance (reflecting ion permeation). Reviewer 3 thinks that important information about the screening is still missing and asks for the rational of the basis of the selection of the library compound, number of compounds tested, number of positive hits detected, stats, etc, among other issues.

In light of the reviews and discussions with the Academic Editor, we would like to invite you to revise the work one last time asking you to thoroughly address all the remaining points raised by the reviewers.

Given the extent of revision needed, we cannot make a decision about publication until we have seen the revised manuscript and your response to the reviewers' comments. Your revised manuscript is likely to be sent for further evaluation by all or a subset of the reviewers.

**IMPORTANT - SUBMITTING YOUR REVISION**

3. Resubmission Checklist

a) *PLOS Data Policy*

b) *Published Peer Review*

c) *Blurb*

Please also provide a blurb which (if accepted) will be included in our weekly and monthly Electronic Table of Contents, sent out to readers of PLOS Biology, and may be used to promote your article in social media. The blurb should be about 30-40 words long and is subject to editorial changes. It should, without exaggeration, entice people to read your manuscript. It should not be redundant with the title and should not contain acronyms or abbreviations. For examples, view our author guidelines: https://journals.plos.org/plosbiology/s/revising-your-manuscript#loc-blurb

Sincerely,

Ines

--

Ines Alvarez-Garcia, PhD

Senior Editor

PLOS Biology

Reviewers' comments

Rev. 2:

I genuinely appreciate the authors' efforts to answer all my concerns. However, I still have problems with the manuscript.

Major concerns:

1. The work is trying to determine the gating mechanism of PAC channels. This is a significant problem that needs to be experimentally addressed. When dealing with gating, it is necessary to show and analyse data related to gating: macroscopic conductance vs voltage or Po vs voltage or macroscopic conductance vs ligand concentration or Po vs ligand concentration. Neither of these are shown in this work. To make the matter more complicated, PAC has a non-Ohmic behaviour. Thus, its macroscopic outward rectification is caused by a variable single-channel conductance; as the membrane potential became positive, the single-channel conductance increased. Therefore, inferring the gating mechanism from the current-voltage relationships or the dose-response curves alone would be difficult.

2. It is still unclear how the C77304 helped the authors infer the gating mechanism. This compound has a bimodal action, which differs from a bidirectional mechanism. It can enhance the macroscopic current in the middle range concentration and inhibit PAC current at high concentrations. The authors propose that C77304 changes the threshold of PAC to respond to protons. By doing so, it is helping channel gating. It is quite possible, but it still is too blurry how C77304 helps.

3. The authors proposed three models and claim that model 1 "fully recapitulates the bidirectional modulation of C77304 on the PAC channel". In any case, it is hard to judge the ability of the proposed models to explain the data without fitting the data (G vs V or Po vs V curves; see point 1) with the equations describing the models. Therefore, my suggestion is to move the mechanistic sections to the discussion and leave there as a hypothetical model that may help to explain the data.

4. The manuscript needs editing. The description of the manuscript is still poor. For example, Figures 1 and 2 are difficult to understand. Punctuation needs revision; it is a particular problem in the Figure legends.

Minor points:

1. The words bi-functional or bi-modal instead of bidirectional would be better descriptors of the effects of C77304 on PAC.

2. The abstract needs revision to include the conclusion of the C77304 experiments.

3. Abstract line 7 needs editing: Furthermore, we revealed a protonation-independent voltage activation of PAC that appear to follow the ion-flux.

4. Page 4, line 7: because most of the channels were in the open state.

5. Protonation-independent voltage gating of PAC instead of noncanonical proton gating-independent voltage gating

6. Page 5, lines 32-33: This is not necessarily true. Lambert and Oberwinkler showed in 2005 that the endogenous PAC channel displayed a strong single-channel conductance outward rectification.

7. Page 5, line 39: increasing pH

8. Page 5, lines 41- 42: Figure 2F shows that the current at +40 and +80 mV is nearly 0 when the extracellular pH is set to 6.0. Therefore, the result is not necessarily surprising.

9. Page 6, line 30: It is hard to rule out the presence/needs of accessory subunits. There are structural examples showing proteins without another protein. For example, the CLC channel's structure did not show Bartin. Hence, at this moment, with the available information, we cannot rule out accessory subunits in PAC channels.

10. Page 7, line 13: acidic-direction shifted threshold activation pHs. Hard to understand; it needs revision.

11. Page 9, line 37: Why is this result intriguing?

12. First line of page 12: mechanism instead of model.

13. Since you are not measuring proton binding, remove the words association and dissociation and describe the values of the time constant of the observations.

14. Page 23, lines 24-25: what exactly do you want to transmit with this sentence?

15. The MD simulations data were gathered at 0 mV. Looking at the IV curves collected at pH 4.0, the size of the current is near 0 at 0 mV, which suggests that the open probability is relatively small. Is it a matter of concern about the mechanistic inferences made with these data?

Rev. 3:

The authors have provided replies to some of the concerns that were raised. In my opinion, the following points still require the authors’ attention:

“We have added details of the screening: “Pharmacological agents interfering with the proton gating of the PAC channel are valuable molecular tools for dissecting its gating mechanism. We screened a compound library (from Selleck Chemicals LLC, Catalog No. L3600) for such agents using manual patch-clamp analysis. Compounds demonstrating >50% inhibition (at 10 µM) of PAC currents in HEK293T cells at pH4.6 were considered as positive ‘hits’ and tested further at a less acidic pH of 5.34 to evaluate any possible pH-dependence of inhibition”, on Page 3, Line 27-32, in this revision.”

Several details will need to be clarified for this part of the study to be suitable for a scientific report. Required information includes, but is not limited to: (i) the rationale at the basis of the selection of the library compound; (ii) the number of compounds that were tested; (iii) the number of positive hits that were detected; (iv) the molecular structure of each individual positive hits and how they compare with C77304; (v) the number of independent experiments and appropriate statistics; (vi) membrane potential at which the recordings were performed etc.

It is also unclear how the C77304 was identified from this analysis given that C77304 does not produce 50% inhibition at 10 uM but only a modest activation (see Fig 1H), while 50% inhibition is the declared threshold for identification of positive hits.

“Yes, the reviewer is right that the C77304 and proton-activated PAC channels at less acidic pHs (pH5.34, pH5.6, and pH5.8) accounted only a small proportion of the total channels available.”

I found the terminology used in the sentence unclear and inaccurate. It is not clear/correct to state that PAC channels at less acidic pHs accounted only for “a small proportion of the total channels available”. It is preferable to state that the open probability of the channel in the pH range mentioned above is rather low. Please ensure accuracy and clarity of the text when referring to this idea.

“As addressed in revised Fig. 2, …...”

In the resubmitted version of the manuscript, there are a series of revised figures including Fig 1 E,F and Fig. 2D. The authors are invited to clarify how the revision of the dataset was performed. Were experiments removed and replaced with new recordings? If this was the case, the exact criteria for how experiments were excluded will need to be declared in the methods. For example, the data set of Fig 2 D changes in appearance somewhat significantly compared to the corresponding data set in the previous version of the ms, but the number of independent experiments was apparently only increased by 2-3 (from 6-7 to 9). Was the addition of 2-3 recordings sufficient to significantly impact on the appearance of the data set or were other criteria (e.g. leak subtraction, data selection) used to obtain the next data set?

“ …new data into the revised Fig. 1H and rewrittent the text as “The proton- and compound-activated channels, however, accounted for a small population of the total available channels (Fig. 1H)” (Page 4, Lines 8-10).”

Fig 1H is missing statistical comparison. It seems essential that this is performed and reported in the figures.

Please also see the comment above about the lack of clarity of the expression “total available channels” in this context.

This figure clearly shows that C77304 is a low potency agonist (and other data sets in the paper show that the molecule is also a weak inhibitor). This should be conveyed in the abstract for clarity; thus lines 4-6 of abstract should read “C77304 acted as low potency agonist and produced moderate activation of the PAC by acting on its proton gating, while simultaneously inhibiting channel activity at much higher concentrations thus also acting as a low potency inhibitor….”

I still think Fig 1G is misleading (please see dedicated comment in the feedback provided in relation to the first submission of the ms) and it should be removed from the figure.

“Additional comments on putative binding sites on the PAC channel”

While the authors provide an interpretation of the data in the context of two separate binding sites, it is appropriate to use the expression “putative binding site” on each occasion these sites are mentioned in the text, since additional direct experimental evidence would be required to unequivocally define the proposed sites of action of C77304. Note that the challenges of separating binding from gating effects have already been largely considered in the field (e.g. doi: 10.1038/sj.bjp.0702164).

---

## [Editor Report · Decision Letter 2]

26 Jul 2023

Dear Dr Tang,

Thank you for your patience while we considered your revised manuscript "A new polymodal gating model of the proton-activated chloride channel" for publication as a Research Article at PLOS Biology. Please note that I am currently handling your manuscript as my colleague Ines is out of the office. This revised version of your manuscript has been evaluated by the PLOS Biology editors and the Academic Editor.

Based on our Academic Editor's assessment of your revision, I am pleased to say that we are likely to accept this manuscript for publication, provided you satisfactorily address the following data and other policy-related requests that I have provided below (A-D):

(A) You may be aware of the PLOS Data Policy, which requires that all data be made available without restriction: http://journals.plos.org/plosbiology/s/data-availability. For more information, please also see this editorial: http://dx.doi.org/10.1371/journal.pbio.1001797

-Supplementary files (e.g., excel). Please ensure that all data files are uploaded as 'Supporting Information' and are invariably referred to (in the manuscript, figure legends, and the Description field when uploading your files) using the following format verbatim: S1 Data, S2 Data, etc. Multiple panels of a single or even several figures can be included as multiple sheets in one excel file that is saved using exactly the following convention: S1_Data.xlsx (using an underscore).

-Deposition in a publicly available repository. Please also provide the accession code or a reviewer link so that we may view your data before publication. 

Figure 1A-L, 2A-F, 3A-M, 4A-H, 5A-F, 6A-E, 6G, 7A-B, S1A-C, S2A-C, S3A-D, S3F, S4A-E, S4G-I, S5A-B, S5D, S6B-I, S7A-O, S8C, S9B-D, S10B

(B) Please also ensure that each of the relevant figure legends in your manuscript include information on *WHERE THE UNDERLYING DATA CAN BE FOUND*, and ensure your supplemental data file/s has a legend.

(C) Please ensure that your Data Statement in the submission system accurately describes where your data can be found and is in final format, as it will be published as written there. 

(D) Please also provide a blurb which (if accepted) will be included in our weekly and monthly Electronic Table of Contents, sent out to readers of PLOS Biology, and may be used to promote your article in social media. The blurb should be about 30-40 words long and is subject to editorial changes. It should, without exaggeration, entice people to read your manuscript. It should not be redundant with the title and should not contain acronyms or abbreviations. For examples, view our author guidelines: https://journals.plos.org/plosbiology/s/revising-your-manuscript#loc-blurb

We expect to receive your revised manuscript within two weeks. 

*Published Peer Review History*

*Press*

Sincerely,

Richard 

Richard Hodge, PhD

rhodge@plos.org

On behalf of:

Ines Alvarez-Garcia, PhD

PLOS

---

## [Editor Report · Decision Letter 3]

23 Aug 2023

Dear Dr Tang,

Thank you for the submission of your revised Research Article entitled "A new polymodal gating model of the proton-activated chloride channel" for publication in PLOS Biology. On behalf of my colleagues and the Academic Editor, Raimund Dutzler, I am delighted to let you know that we can in principle accept your manuscript for publication, provided you address any remaining formatting and reporting issues. These will be detailed in an email you should receive within 2-3 business days from our colleagues in the journal operations team; no action is required from you until then. Please note that we will not be able to formally accept your manuscript and schedule it for publication until you have completed any requested changes.

PRESS

Sincerely, 

Ines

--

Ines Alvarez-Garcia, PhD

Senior Editor

PLOS Biology
